# Light as Deception: GPT-driven Natural Relighting Against Vision-Language Pre-training Models

## Abstract

While adversarial attacks on vision-and-language pretraining (VLP) models have been explored, generating natural adversarial samples crafted through realistic and semantically meaningful perturbations remains an open challenge. Existing methods, primarily designed for classification tasks, struggle when adapted to VLP models due to their restricted optimization spaces, leading to ineffective attacks or unnatural artifacts. To address this, we propose **LightD**, a novel framework that generates natural adversarial samples for VLP models via semantically guided relighting. Specifically, LightD leverages ChatGPT to propose context-aware initial lighting parameters and integrates a pretrained relighting model (IC-light) to enable diverse lighting adjustments. LightD expands the optimization space while ensuring perturbations align with scene semantics. Additionally, gradient-based optimization is applied to the reference lighting image to further enhance attack effectiveness while maintaining visual naturalness. The effectiveness and superiority of the proposed LightD have been demonstrated across various VLP models in tasks such as image captioning and visual question answering.

## 1. Introduction

Vision-and-language pre-training (VLP) models have significantly advanced the integration of visual and language modalities by leveraging large-scale image-text datasets and multimodal learning techniques. With increases in data volume, model parameters, and computational power, these VLP models have achieved notable success and demonstrated impressive capabilities across various downstream vision-and-language (V+L) tasks, such as image captioning

[1]Anonymous Institution, Anonymous City, Anonymous Region, Anonymous Country. Correspondence to: Anonymous Author <anon.email@domain.com>.

Preliminary work. Under review by the International Conference on Machine Learning (ICML). Do not distribute.

and visual question answering (VQA) (Chen et al., 2022; Alayrac et al., 2022; Tsimpoukelli et al., 2021; Gupta et al., 2022). Models like CLIPCap (Mokady et al., 2021), BLIP (Li et al., 2022), BLIP2 (Li et al., 2023a), and Image2LLM (Guo et al., 2023) have shown exceptional results in these areas. Nevertheless, recent studies have revealed the vulnerability of VLP models to adversarial attacks (Zhang et al., 2022; He et al., 2023; Han et al., 2023; Lu et al., 2023; Cheng et al., 2024; Gao et al., 2025).

However, all these attacks primarily focus on adding human-imperceptible perturbations to clean images within $L_p$-norm constraints. Although these attacks are effective in certain scenarios, they are typically susceptible to adversarial denoising techniques, limiting their practice (Xie et al., 2019). To address these limitations, "non-suspicious" adversarial attacks have emerged as a more realistic threat. These attacks allow for subtle yet unrestricted modifications, such as color adjustments (Hosseini & Poovendran, 2018; Shamsabadi et al., 2020b; 2021; Zhao et al., 2023), lighting changes (Shamsabadi et al., 2020a; Gao et al., 2022; Huang et al., 2023; Zhang et al., 2024), and semantic alterations (Joshi et al., 2019). Although these methods have shown promise in image classification tasks, their effectiveness and robustness against VLP models remain underexplored.

In this study, we first investigate the robustness of VLP models against current non-suspicious adversarial attacks. Specifically, we introduce a general optimization framework for adapting existing non-suspicious adversarial attacks from image classification tasks to VLP models for downstream V+L tasks. These attacks primarily adjust parameters related to semantic characteristics (such as lighting and color) based on the optimization objection, or introduce perturbations directly to the generated adversarial images, resulting in the victim models outputting erroneous predictions. Unfortunately, these methods fail to achieve adversarial attack performance and visual naturalness simultaneously. (See Section 5.2 for more details.)

To address this challenge, we propose **LightD** (Light as Deception), a novel GPT-driven adversarial relighting framework designed to deceive VLP models by adjusting the lighting of clean images. LightD comprises three key components: GPT-based lighting parameter selection, relighting-

driven adversarial image generation, and two-step collaboration optimization process. Specifically, LightD utilizes the pre-trained relighting model IC-Light (Zhang et al., 2025) to apply consistent lighting conditions to the clean image. To generate an appropriate reference lighting image for each clean image, we adopt ChatGPT (gpt-4o-2024-08-06) to determine the initial lighting parameters. Furthermore, we propose a lighting-based collaboration optimization strategy to create efficient adversarial relighted images. Such a strategy enables LightD to achieve a balance of attack performance and visual naturalness by adjusting lighting parameters and adding corruptions to the reference lighting image. The main contributions can be summarized as:

- We propose a general framework for transferring non-suspicious adversarial attacks from image classification tasks to VLP models, revealing that these attacks struggle to balance effectiveness with visual naturalness.

- We introduce **LightD**, a novel GPT-driven adversarial relighting technique against VLP models. Our approach is enhanced through GPT-based initial point selection and SGA-based collaborative optimization, which introduce perturbations to lighting parameters and the reference lighting image.

- Extensive experiments verify the superior efficacy of LightD in attacking VLP models on image captioning and VQA while maintaining high visual naturalness.

## 2. Related Works

### 2.1. VLP Models and Their Robustness

Vision-and-language pre-training aims to enhance the performance of subsequent multimodal tasks by pre-training on extensive image-text pairs. Based on their architectures, VLP models can be categorized into fused and aligned models (Zhang et al., 2022). In fused VLP models (e.g., TCL (Yang et al., 2022), ALBEF (Li et al., 2021a), BLIP (Li et al., 2022)), image and text information are integrated into a shared and unified representation space. Typically, a joint encoder (such as a multimodal Transformer) is used to simultaneously process and integrate multimodal information. Conversely, aligned VLP models (e.g., CLIP (Radford et al., 2021)) process image and text information through separate encoders. After being encoded separately, these two modalities' representations are aligned or associated using a certain alignment mechanism, such as contrastive learning or matching loss. Inspired by the adversarial vulnerability observed in vision and language tasks, early research has focused on investigating adversarial attacks against VLP models in fields such as image-text retrieval (Zhang et al., 2022; Gao et al., 2025), image captioning (Xu et al., 2018; 2019; Ji et al., 2020; Aafaq et al., 2021; Li et al., 2024), and

VQA (Xu et al., 2018; Li et al., 2021b; Sheng et al., 2021; Cao et al., 2022). Adversarial attacks can be categorized into white-box and black-box attacks (Gu et al., 2023). White-box attacks (Jia et al., 2024) have full access and knowledge of the model, whereas black-box attacks (Park et al., 2024; Bai et al., 2020) do not and are more representative of actual application scenarios. Most of these studies have concentrated on traditional CNN-RNN-based models, assuming white-box access or untargeted adversarial objectives, and requiring human intervention.

### 2.2. Non-suspicious Adversarial Attacks

Adversarial attacks have achieved a remarkable ability to deceive well-trained deep-learning models across various applications (Xie et al., 2017; Dong et al., 2020; Li et al., 2023b). Traditionally, these attacks have been developed under the premise that adversarial examples should be indistinguishable from their corresponding clean images, often achieved by optimizing with $L_p$-norm constraints (Carlini & Wagner, 2017). However, recent research has challenged this assumption, arguing that it lacks practical relevance in real-world scenarios (Gilmer et al., 2018). Since there is no direct comparison with the original image, adversarial images can remain inconspicuous without strictly limiting the perturbations. Thereafter, numerous non-suspicious adversarial attacks target domain-specific attributes have been explored, including light changes (Shamsabadi et al., 2020a; Gao et al., 2022; Huang et al., 2023; Zhang et al., 2024), color adjustments (Hosseini & Poovendran, 2018; Shamsabadi et al., 2020b; 2021; Zhao et al., 2023), and semantic alterations (Joshi et al., 2019). These attacks mainly focus on adjusting parameters related to semantic characteristics via the optimization objection. However, the adversarial images of these non-suspicious attacks cannot satisfy both attack performance and visual naturalness for VLP models (See Section 5.2).

### 2.3. Downstream Vison and Language Tasks

**Image Captioning** is a multimodal task that combines computer vision and natural language processing to generate descriptive captions for images. The process typically involves extracting visual features from an image using a CNN and then utilizing these features to produce a caption through a language model, often implemented as an RNN or a Transformer (Donahue et al., 2015; Fang et al., 2015; Huang et al., 2019). Advanced image captioning models employ encoder-decoder architectures with attention mechanisms to enhance their ability to focus on the most relevant parts of the image (Chen et al., 2017; Anderson et al., 2018). These models are trained on large datasets of images and their corresponding captions, learning to map visual content to textual descriptions.

**Visual Question-Answering** requires models to understand both visual and linguistic information to answer questions about images. Early VQA models were inspired by image captioning approaches, using CNNs for image encoding and RNNs for question encoding (Malinowski et al., 2015; Gao et al., 2015). However, the field has evolved significantly, with the introduction of attention mechanisms that allow models to focus on specific regions of an image while answering a question (Lu et al., 2016; Shih et al., 2016; Sood et al., 2023). Recently, Transformer-based models have achieved SOTA performance in VQA through large-scale pre-training on visual and linguistic data (Tan & Bansal, 2019; Zhang et al., 2021). These models leverage the self-attention mechanism of Transformers to capture complex interactions between visual and linguistic features, enabling them to answer a wide range of questions about images.

## 3. Preliminary

**Problem Formulation.** Adversarial attacks on VLP models involve creating discrepancies between perturbed images and their corresponding texts, while adhering to predefined limitations on the perturbations. Let $(I, T)$ denote an image-text pair, and let $I'$ be the corresponding adversarial counterpart. This paper focuses on non-suspicious adversarial attacks that may not be constrained by slight changes to images. The image will be modified by any transformation, provided that the transformation preserves the visual semantic content of the image. Let $\Omega$ represent the human visual system (HVS), the problem of crafting non-suspicious adversarial attack on VLP models can be formulated as:

$$I' = \underset{\Omega(I')=\Omega(I)}{arg\ max}\ \mathbf{J}\left(f_\phi(I'), f_\varphi(T)\right) \tag{1}$$

where $f_\phi$ and $f_\varphi$ represent the image encoder and text encoder of the multimodal model, respectively; $\mathbf{J}$ rates the cross-modality similarity of $I'$ and $T$.

**Research Gaps.** Non-suspicious adversarial attacks have been extensively studied and have achieved significant success in deep-learning models. Until now, research on their effects on VLP models is still underexplored. As we know, we are the first work to investigate the effectiveness of non-suspicious adversarial attacks against VLP models. This study addresses the gap from two aspects. First, we develop a general optimization objective that allows existing non-suspicious adversarial attacks for image classification to be adapted to VLP models for downstream V+L tasks. Since these attacks usually optimize the semantic parameters related to lighting and color without constraints, they cannot obtain promising performance in terms of attack performance and visual naturalness simultaneously. Second, we propose a novel non-suspicious adversarial relighting attack tailored for VLP models according to a pre-trained relighting model IC-Light (Zhang et al., 2025).

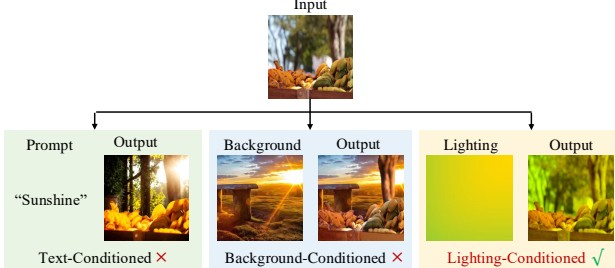

**Figure 1:** The first two columns: the default lighting types provided in IC-Light (Zhang et al., 2025). The last column: lighting strategy used in our paper.

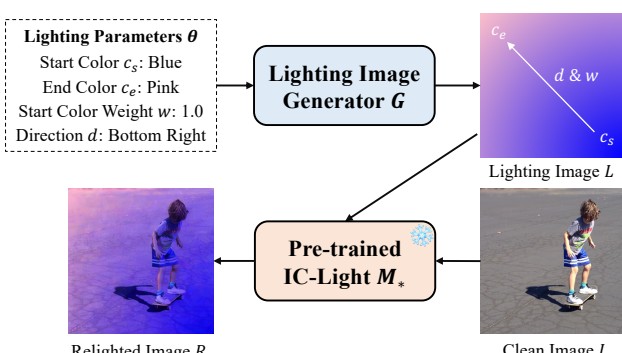

**Figure 2:** The whole relighting procedure consists of 1) generation of the reference lighting image by $G$ (ComfyUI, 2024) and 2) the subsequent relighting via IC-Light model (Zhang et al., 2025).

**IC-Light.** IC-Light (Zhang et al., 2025) is a diffusion-based relighting model to impose consistent lighting on images, ensuring precise illumination modification while preserving intrinsic image details. The key innovation of IC-Light lies in its ability to leverage the property of illumination independence in HDR space, ensuring that the blending of appearances from different light sources results in a mathematically equivalent appearance with mixed light sources. This consistency is enforced using multi-layer perceptions (MLPs) in latent space during model training, enabling the production of highly coherent and realistic relighting effects. IC-Light supports two forms of lighting modification by default: ❶ Text-conditioned method: adjusts the illumination of clean samples through illumination-related text instructions. ❷ Background-conditioned method: uses a reference background image to introduce its illumination information into the clean sample. As shown in Fig. 1, these two methods result in semantic differences, especially in the background areas between the relighted and the clean images, making them unsuitable for adversarial attacks on VLP models. To address this issue, this paper proposes to use lighting-conditioned solution, leveraging a pure lighting image as a reference under the background-based method. The output relighted images are with only illumination differences and no semantic content differences (see the last column of Fig. 1).

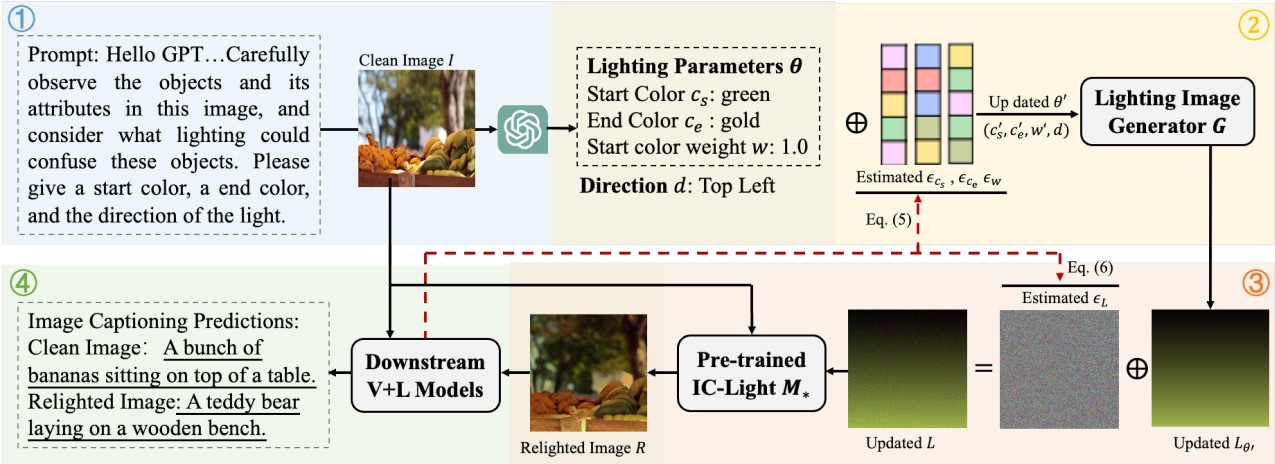

**Figure 3:** Overview of the proposed **LightD**. (1) We use the ChatGPT to accommodate the initial lighting parameters (*i.e.*, start color, end color, and direction) for (2) reference lighting image generation; (3) Collaboration optimization for adversarial relighted image based on pre-trained IC-Light model; (4) Attack VLP models (*e.g*., CLIPCap for image captioning tasks).

In a nutshell, this paper tackles three main challenges: 1) Generating pure lighting images as reference background for relighting. We leverage a lighting image generation function **G** proposed in (ComfyUI, 2024) to create the reference lighting image $L$ by taking the parameters start color $c_s$, end color $c_e$, weight $w$, and light direction $d$ as input, where $w$ denotes the proportion of $c_s$ in the entire interval of $L$, ranging in [0, 2]. Fig. 2 illustrates the process of pure lighting image generation. 2) Ensuring the naturalness of the relighted image. We attempt to use GPT as a recommender to adaptively select the appropriate parameters (color, direction, and weight) for lighting image generation by analyzing the visual content of the clean image. 3) Ensuring attack performance against VLP models. We propose a lighting-cooperation optimization strategy. This strategy first optimizes the recommended parameters and then optimizes the generated reference lighting image to improve attack performance, which enlarges the optimized space compared with the baseline methods.

## 4. Method

### 4.1. Overview

Fig. 3 illustrates the framework of our proposed GPT-driven natural adversarial relighting attack **LigthD**. Our goal is to deceive the VLP models based on a pre-trained image relighting model IC-Light (Zhang et al., 2025), thereby segmenting into two distinct phases: the relighting phase and the attack phase. During the relighting phase, ① we harness the formidable reasoning prowess of ChatGPT to ascertain appropriate initial lighting parameters, such as colors and direction. Then, ② we leverage lighting image generator to produce the initial reference lighting image. Subsequently, we can deploy the pre-trained relighting model IC-Light

(Zhang et al., 2025) to generate the initial adversarial relighting image. Transitioning to the attack phase, ③ & ④ we build upon the optimization idea of SGA (Lu et al., 2023) to fine-tune both lighting parameters and reference lighting image, meticulously crafting adversarial relighted images to satisfy visual naturalness and attack performance. LightD comprises three pivotal design elements: adoptive GPT-based lighting parameter selection, relighting-driven adversarial image generation, and SGA-guided collaborative optimization. We describe each module in details below.

### 4.2. GPT-driven Relighted Image Generation

**GPT-Based Lighting Parameter Selection.** To ensure that the relighted images align more closely with typical visual perception, we adopt ChatGPT to suggest initial lighting colors and direction. As depicted in Fig. 3, we design a prompt template that guides ChatGPT in generating the lighting parameters $\theta$ (start and end colors $c_s$, $c_e$) and lighting direction $d$. More details of the prompt template can be found in Appendix B.

**Lighting Image Generation.** Once we have acquired the initial lighting parameters $\theta$, including the start color $c_s$ with weight $w$, end color $c_e$, and lighting direction $d$, we create a reference lighting image that guides the subsequent relighting process. By utilizing these parameters accommodated by ChatGPT , the generated reference lighting image aligns with the desired aesthetic and functional requirements, setting the stage for the creation of natural adversarial relighted images. The reference lighting image $L$ is represented as:

$$L = \mathbf{G}(c_s, c_e, d, w), \tag{2}$$

where **G** is the lighting image generation function provided in (ComfyUI, 2024).

**Adversarial Relighted Image Generation.** After successfully acquiring the initial reference lighting image $L$, we harness the pre-trained IC-Light (Zhang et al., 2025) to produce the adversarial relighted image $R$. Essentially, $R$ retains the basic visual content of the clean image while introducing subtle yet significant changes in illumination. By applying the lighting effect of $L$, we can obtain $R$:

$$R = \mathbf{M}_*(L, I), \tag{3}$$

where $\mathbf{M}_*$ is the pre-trained relighting model IC-Light. According to the powerful generation ability of the diffusion model, the IC-Light allows imposing consistent light from the reference lighting image into the original clean image without structure and visual content variation.

### 4.3. Lighting-Based Collaboration Optimization

The primary objective of the proposed method is to create adversarial relighted images that effectively deceive VLP models without causing noticeable artifacts. To achieve this, we introduce a two-step optimization strategy that leverages the optimization idea of SGA (Lu et al., 2023) to iteratively refine both the lighting parameters and the reference lighting image used by the IC-Light model. More details of SGA can be found in Appendix C. In the first step, we focus on optimizing the lighting parameters. It is devoted to adjusting the parameters that maximize the confusion of the VLP models. In the second step, we optimize the reference lighting image based on the optimal lighting parameters from the first step. By iteratively refining lighting parameters and reference lighting image, LightD can achieve visual naturalness and high attack performance.

**Lighting Parameter Optimization.** The objection of lighting parameters optimization is to identify the optimal set of lighting parameters $\theta$ (start color $c_s$, initial weight $w = 1.0$, and end color $c_e$). Specifically, we utilize the following loss function to obtain the optimal parameter perturbations $\epsilon_\theta$:

$$\epsilon_\theta = \underset{\Omega(R)=\Omega(I)}{arg\,max}\ \mathbf{J}(f_\phi(\mathbf{M}_*(\mathbf{G}(\theta), I)), f_\varphi(T)), \tag{4}$$

where $R = \mathbf{M}_*(\mathbf{G}(\theta), I))$ denotes the relighted image; $\mathbf{J}$ denotes the objective loss function; $f_\phi$ and $f_\varphi$ are the image and text encoders of VLP models, respectively. Let $\theta_i$ denote the lighting parameters at the $i$th step, the parameter iteration process can be formulated as:

$$\theta_{i+1} = \theta_i + \alpha \cdot sign \frac{\bigtriangledown_{\theta_i}\mathbf{J}(f_\phi(R_{\theta_i}), f_\varphi(T))}{||\bigtriangledown_{\theta_i}\mathbf{J}(f_\phi(R_{\theta_i}), f_\varphi(T))||}, \tag{5}$$

where $R_{\theta_i} = \mathbf{M}_*(L_{\theta_i}, I)$ denotes the generated adversarial relighting image under the lighting parameters $\theta_i$, $L_{\theta_i} = \mathbf{G}(\theta_i)$ denotes the reference lighting image at the $i$th step.

**Reference Lighting Image Optimization.** Once the optimal lighting parameters have been identified, the next step

in our optimization process is to refine the reference lighting image. The goal of this optimization step is to further enhance the effectiveness and robustness of the adversarial examples generated by the model. To achieve this, we leverage the core idea of the SGA (Lu et al., 2023) to optimize Eq. 4 by enhancing diversity. Let $L_i$ denotes the generated reference lighting image at the $i$th step, we first resize $L_i$ $M$ times and obtain a expand set $\{L_{i1}, L_{i2}, \ldots, L_{iM}\}$. Then the expanded set and the clean image are fed into the pre-trained relighting model, obtaining $M$ relighting images $\{R_{i1}, R_{i2}, \ldots, R_{iM}\}$, where $R_{ij} = \mathbf{M}_*(L_{ij}, I)$, $j = 1, 2, \ldots, M$. The iteration process is formulated as:

$$L_{i+1} = L_i + \alpha \cdot sign \frac{(\bigtriangledown_L \sum_{j=1}^M \mathbf{J}(f_\phi(R_{ij}), f_\varphi(T)))}{||\bigtriangledown_L \sum_{j=1}^M \mathbf{J}(f_\phi(R_{ij}), f_\varphi(T))||}, \tag{6}$$

where $R_{iM} = \mathbf{M}_*(L_{iM}, I)$ is the generated adversarial relighting image. After obtaining the optimal reference lighting image, we employ the pre-trained IC-Light (Zhang et al., 2025) to generate the adversarial relighted images.

The number of resize times is critical for the model's performance, we provide the comparison of different sizes in Appendix E.

**Loss Function of Optimization.** The loss function $\mathbf{J}$ is crucial for optimizing both the lighting parameters and the reference lighting image in our proposed method. To achieve the desired objectives of generating adversarial relighting samples that are both effective and visually natural, we design the loss function to balance two key conditions. (1) *Attack capability*: the encoding of the adversarial relighting image should be as different as possible from the encoding of the label text. This ensures that the adversarial image has a high likelihood of deceiving the target VLP models, causing it to produce incorrect text recognitions. (2) *Visual naturalness*: the encoding of the adversarial relighting image should be as similar as possible to the encoding of the original clean image. This ensures that the adversarial image retains the visual characteristics of the original image, making it less detectable as an adversarial example. Given these conditions, the loss function $\mathbf{J}$ for lighting parameter and lighting image optimization can be defined as follows:

$$\mathbf{J} = arg\,max(CE(f_\phi(R), f_\varphi(T)) + CS(h_\phi(R), h_\phi(I))), \tag{7}$$

where $CE$ denotes the loss function (e.g., cross-entropy loss) of the victim model, $f_\phi$ and $f_\varphi$ are the image encoder and text encoder of the victim model, respectively; $CS$ denotes the cosine similarity loss, $h_\phi$ is the CLIP image encoder, we select the ViT-B/32 version of CLIP here.

### 4.4. Transfer Non-suspicious Attacks to V+L Tasks

Existing natural non-suspicious adversarial attack methods, such as adversarial relighting attacks and adversarial color

attacks, have primarily been designed for image classification tasks. These methods cannot be directly applied to VLP models due to different objections. In this study, we propose a general strategy to transfer these non-suspicious adversarial attack methods from image classification tasks to V+L tasks. Let $(I, l)$ denote a pair of a clean image and its corresponding text label, $I'$ denotes the adversarial image generated by the attack. In image classification tasks, the termination condition for the optimization iteration process of an adversarial attack is typically $\mathbf{F}(I') \neq l = \mathbf{F}(I)$, where $\mathbf{F}$ denotes the victim model. To adapt these attacks for VLP models, we develop a new termination condition that maximizes the specifically designed loss function $\mathbf{J}$ in Eq. 7. This loss function is tailored to the objectives of the V+L tasks, incorporating both the attack capability and visual naturalness of the generated adversarial images.

## 5. Experiments

### 5.1. Setups

**Datasets.** In this study, we verify the effectiveness of our LightD against open-source VLP models on two typical downstream V+L tasks: image captioning and VQA. To achieve this, we leverage three widely used multimodal image captioning datasets: MSCOCO (Lin et al., 2014), Flickr8K (Hodosh et al., 2013), and Flickr30K (Plummer et al., 2015). For the VQA task, we employ the MSCOCO and DAQUAE (Malinowski & Fritz, 2014) datasets. We randomly select 1,000 images from the test set of each dataset to serve as the clean images for adversarial generation.

**Baseline Methods.** We compare the proposed method with SOTA non-suspicious adversarial attack methods. They are three adversarial relighting attacks ALA (Huang et al., 2023), EdgeFool (Shamsabadi et al., 2020a), and Jadena (Gao et al., 2022), and three adversarial color attacks SemanticAdv (Hosseini & Poovendran, 2018), ColorFool (Shamsabadi et al., 2020b), and AdvCF (Zhao et al., 2023).

**Victim VLP Models.** We employ several typical VLP models for different downstream V+L tasks to demonstrate the effectiveness of the proposed method. Specifically, we use CLIPCap (Mokady et al., 2021), BLIP (Li et al., 2022), and BLIP2 (Li et al., 2023a) for image captioning and BLIP and BLIP2 are used for VQA.

**Evaluation Metrics.** Image captioning typically utilizes BLEU (Naseer et al., 2021), METEOR (Banerjee & Lavie, 2005), ROUGE (Chin-Yew, 2004), CIDEr (Vedantam et al., 2015), and SPICE (Anderson et al., 2016) to assess the quality and relevance between the predicted and reference captions. For the VQA task, the average prediction accuracy (APA) and WUPS0.9 (Kafle & Kanan, 2017) are used to measure model's performance. We employ a no-reference image quality index to assess the naturalness of the gener-

**Table 1:** Comparison with state-of-the-art methods for image captioning task on MSCOCO and Flickr30K datasets.

| Dataset | Model | Attack | BLEU↓ | METEOR↓ | ROUGE_L↓ | CIDEr↓ | SPICE↓ | NIQE↓ |
|---|---|---|---|---|---|---|---|---|
| MSCOCO | CLIPCap | SemanticAdv | 0.538 | 0.179 | 0.418 | 0.463 | 0.106 | 9.479 |
| | | ColorFool | 0.584 | 0.199 | 0.448 | 0.590 | 0.126 | 9.679 |
| | | AdvCF | 0.519 | 0.164 | 0.395 | 0.390 | 0.092 | 9.793 |
| | | EdgeFool | 0.463 | 0.124 | 0.351 | **0.200** | **0.052** | 20.003 |
| | | ALA | 0.657 | 0.229 | 0.487 | 0.839 | 0.114 | 9.811 |
| | | Jadena | 0.590 | 0.187 | 0.436 | 0.582 | 0.114 | 20.394 |
| | | **LightD(Ours)** | **0.460** | **0.119** | **0.340** | 0.211 | 0.055 | **8.650** |
| | BLIP | SemanticAdv | 0.642 | 0.237 | 0.493 | 0.840 | 0.167 | **5.672** |
| | | ColorFool | 0.689 | 0.251 | 0.522 | 0.961 | 0.185 | 5.830 |
| | | AdvCF | 0.671 | 0.247 | 0.509 | 0.886 | 0.177 | 5.822 |
| | | EdgeFool | 0.598 | 0.200 | 0.449 | 0.619 | 0.130 | 8.314 |
| | | ALA | 0.762 | 0.293 | 0.572 | 1.251 | 0.222 | 5.793 |
| | | Jadena | 0.703 | 0.257 | 0.525 | 1.021 | 0.187 | 9.913 |
| | | **LightD(Ours)** | **0.554** | **0.177** | **0.419** | **0.502** | **0.110** | 9.783 |
| | BLIP2 | SemanticAdv | 0.638 | 0.220 | 0.496 | 0.835 | 0.167 | 9.308 |
| | | ColorFool | 0.665 | 0.236 | 0.527 | 0.979 | 0.179 | 9.711 |
| | | AdvCF | 0.619 | 0.218 | 0.498 | 0.850 | 0.165 | 9.885 |
| | | EdgeFool | 0.626 | 0.289 | 0.558 | 0.899 | 0.179 | 11.974 |
| | | ALA | 0.713 | 0.260 | 0.562 | 1.178 | 0.207 | 9.651 |
| | | Jadena | 0.628 | 0.220 | 0.509 | 0.927 | 0.168 | 19.819 |
| | | **LightD(Ours)** | **0.605** | **0.204** | **0.483** | **0.811** | **0.156** | **8.352** |
| Flickr30K | CLIPCap | SemanticAdv | 0.495 | 0.138 | 0.358 | 0.172 | 0.075 | 9.663 |
| | | ColorFool | 0.538 | 0.152 | 0.382 | 0.235 | 0.086 | 9.861 |
| | | AdvCF | 0.477 | 0.127 | 0.340 | 0.148 | 0.065 | 10.088 |
| | | EdgeFool | 0.454 | 0.104 | 0.315 | 0.096 | **0.043** | 18.772 |
| | | ALA | 0.573 | 0.162 | 0.395 | 0.301 | 0.097 | 10.339 |
| | | Jadena | 0.539 | 0.142 | 0.368 | 0.228 | 0.077 | 20.139 |
| | | **LightD(Ours)** | **0.430** | **0.097** | **0.306** | **0.086** | **0.043** | **8.409** |
| | BLIP | SemanticAdv | 0.573 | 0.172 | 0.402 | 0.357 | 0.108 | **5.542** |
| | | ColorFool | 0.606 | 0.182 | 0.425 | 0.424 | 0.120 | 5.608 |
| | | AdvCF | 0.597 | 0.178 | 0.416 | 0.389 | 0.115 | 5.703 |
| | | EdgeFool | 0.540 | 0.140 | 0.361 | 0.237 | 0.076 | 9.086 |
| | | ALA | 0.673 | 0.212 | 0.466 | 0.611 | 0.148 | 5.683 |
| | | Jadena | 0.615 | 0.182 | 0.421 | 0.435 | 0.118 | 9.601 |
| | | **LightD(Ours)** | **0.485** | **0.126** | **0.353** | **0.190** | **0.071** | 9.823 |
| | BLIP2 | SemanticAdv | 0.604 | 0.177 | 0.435 | 0.434 | 0.118 | 9.975 |
| | | ColorFool | 0.646 | 0.191 | 0.460 | 0.510 | 0.129 | 10.056 |
| | | AdvCF | 0.603 | 0.176 | 0.437 | 0.442 | 0.117 | 10.371 |
| | | EdgeFool | 0.676 | 0.209 | 0.494 | 0.632 | 0.145 | 14.563 |
| | | ALA | 0.666 | 0.208 | 0.492 | 0.620 | 0.145 | 10.230 |
| | | Jadena | 0.618 | 0.183 | 0.455 | 0.495 | 0.119 | 19.491 |
| | | **LightD(Ours)** | **0.586** | **0.158** | **0.423** | **0.417** | **0.107** | **8.575** |

ated adversarial images, i.e., NIQE (Mittal et al., 2012). A lower NIQE score suggests better image quality.

More details are provided in the Appendix D.

### 5.2. Performance Evaluation

To test the effectiveness of our method, we compare it with SOTA non-suspicious attacks against typical VLP models for image captioning and VQA tasks. For the image captioning task, the comparison is conducted against CLIPCap (Mokady et al., 2021), BLIP (Li et al., 2022), and BLIP2 (Li et al., 2023a) models on MSCOCO, Flickr8K, and Flickr30K datasets. For the VQA tasks, the comparison is conducted on BLIP (Li et al., 2022) and BLIP2 (Li et al., 2023a) models on MSCOCO and DAQUAR datasets.

**Performance on Image Captioning.** Table 1 provides the quantitative comparison results of all methods on MSCOCO and Flickr30K, while the result on Flickr8K is given in the Appendix F. Table 1 shows that the compared non-suspicious attacks have a certain degree of attack capability on these VLP models for image captioning tasks. Such results demonstrate the usefulness of the proposed general optimization framework for transferring these baselines to VLP models in Section 4.4. The proposed method significantly outperforms these baselines across the three victim VLP models on both MSCOCO and Flickr30K datasets,

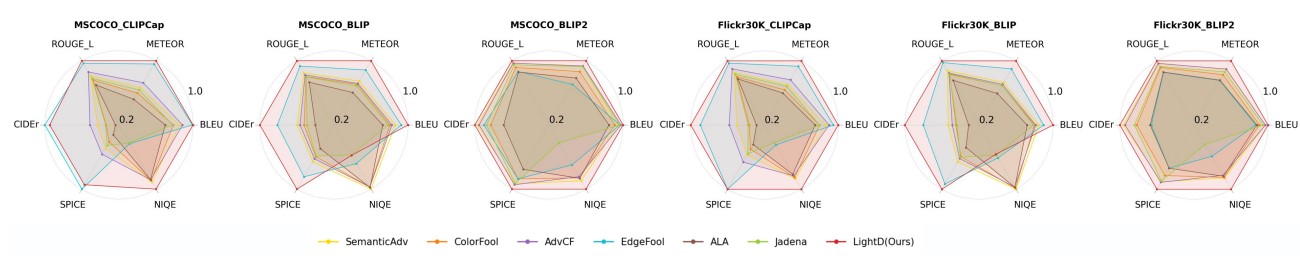

**Figure 4:** Radar chart to illustrate the comparison with state-of-the-art methods for image captioning task on MSCOCO and Flickr30K datasets. Since a lower value of each evaluation metric denotes better attack performance and visual naturalness, the radar charts are computed on normalized values (1/each metric). Thus, a larger region denotes better performance.

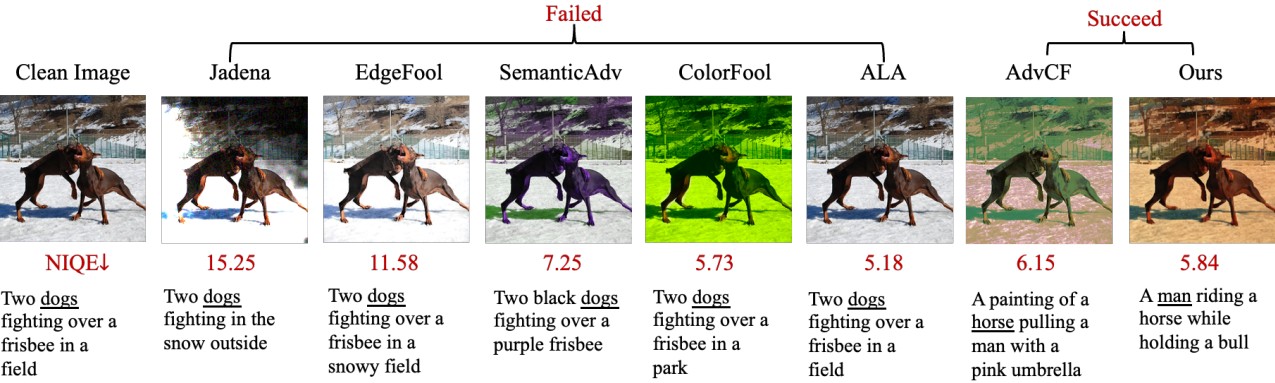

**Figure 5:** Visualization of adversarial examples of attacking ClIPCap model on MSCOCO in image captioning task.

further underscoring its potency. In addition, image quality index NIQE of our method obtains the lowest values among most models on two datasets, which further verifies the naturalness of our generated adversarial relighted images. Furthermore, Fig. 4 provides a more obvious comparison in terms of attack performance and visual naturalness for all methods in image captioning tasks.

To provide a detailed demonstration of our method's advantages, we present adversarial examples and predicted captions for all methods in Fig. 5. These results are obtained by attacking the ClIPCap model using the MSCOCO dataset. It is evident that the adversarial images relighted by our method can effectively deceive models without compromising their natural visual appearance. In contrast, the adversarial examples generated by other methods either fail to deceive the model or exhibit noticeable corruptions. In summary, LightD outperforms state-of-the-art non-suspicious attack methods in achieving a superior balance between attack effectiveness and visual naturalness.

**Performance Comparison on VQA Task.** Table 2 provides the quantitative comparison results of all involved attack methods. It is observed that the compared non-suspicious adversarial attack methods cannot achieve higher attack performance and better visual naturalness simultaneously. In contrast, the proposed lightD performs the best attack performance across two VLP models on both datasets while obtaining lower NIQE values. It reveals the effectiveness

of our method based on the pre-trained relighting model to impose natural and consistent light for clean images. Specifically, the specially designed ChatGPT-based lighting parameter selection and lighting-based collaboration optimization strategy enable the generated relighted images to possess non-suspicious visual perception while preserving the capability to deceive VLP models.

We also provide some visual examples to illustrate VQA results of all methods in Fig. 6. It is observed that the baselines mislead VLP models to predict error answers usually have poor visual quality. These adversarial images, characterized by unnatural colors and corruptions, can be readily identified by human beings. In contrast, the adversarial samples generated by our method can successfully mislead the VLP models while ensuring natural visual perception.

### 5.3. Ablation Study

Our proposed LightD comtains two key components: GPT-driven lighting parameter selection and two-step collaboration optimization. To investigate the impact of each component on the effectiveness of our method, we conduct ablation studies against the BLIP2 model on the MSCOCO dataset in both the image captioning and VQA tasks.

**Evaluation of ChatGPT Recommendation.** To validate the effectiveness of the initial lighting parameters (start color and end color) recommended by ChatGPT , we compare

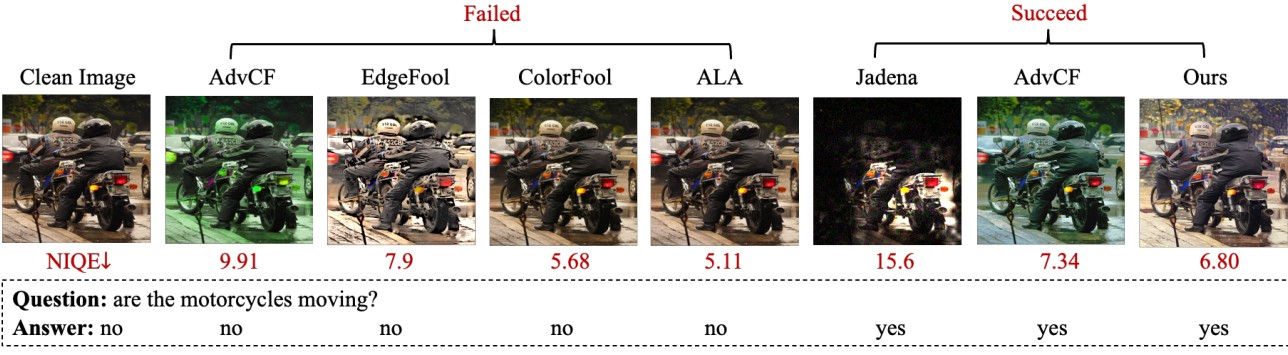

**Figure 6:** Visualization of adversarial examples of attacking BLIP model on MSCOCO in VQA task.

**Table 2:** Performance comparison in VQA task.

| Model | Attack | MSCOCO | | | DAQUAR | | |
|---|---|---|---|---|---|---|---|
| | | APA↓ | WUPS0.9↓ | NIQE↓ | APA↓ | WUPS0.9↓ | NIQE↓ |
| BLIP | SemanticAdv | 61.15 | 0.679 | **5.660** | 12.24 | 0.201 | 9.999 |
| | ColorFool | 68.75 | 0.758 | 5.798 | 15.53 | 0.241 | 9.878 |
| | AdvCF | 62.18 | 0.702 | 5.781 | 12.66 | 0.210 | 9.342 |
| | EdgeFool | 58.54 | 0.668 | 7.584 | 19.78 | 0.260 | 10.478 |
| | ALA | 79.83 | 0.864 | 5.949 | 18.37 | 0.255 | 8.956 |
| | Jadena | 75.18 | 0.827 | 9.902 | 18.09 | 0.257 | 10.554 |
| | **LightD(Ours)** | **58.26** | **0.651** | 5.720 | **11.68** | **0.207** | **8.907** |
| BLIP2 | SemanticAdv | 42.90 | 0.460 | 9.389 | 6.90 | 0.063 | 8.608 |
| | ColorFool | 43.95 | 0.474 | 9.717 | 8.69 | 0.077 | 8.824 |
| | AdvCF | 44.32 | 0.489 | 18.112 | 8.91 | **0.074** | 8.941 |
| | EdgeFool | 51.09 | 0.540 | 13.389 | 14.21 | 0.102 | **8.220** |
| | ALA | 51.11 | 0.541 | 10.081 | 11.96 | 0.098 | 8.761 |
| | Jadena | 48.45 | 0.523 | 19.959 | 12.36 | 0.089 | 23.782 |
| | **LightD(Ours)** | **40.31** | **0.453** | **8.240** | **8.52** | **0.074** | 8.390 |

**Table 3:** Impact of GPT-based initial lighting parameters. We calculate average results over 1000 samples.

| Optimization | BLEU↓ | METEOR↓ | ROUGE_L↓ | CIDEr↓ | SPICE↓ | NIQE↓ |
|---|---|---|---|---|---|---|
| Random | 0.609 | 0.207 | 0.487 | 0.829 | 0.159 | 8.475 |
| GPT | **0.605** | **0.204** | **0.483** | **0.811** | **0.156** | **8.352** |

**Table 4:** Impact of two-step optimizations. We calculate average results over 1000 samples.

| LightPara | LightImg | BLEU↓ | METEOR↓ | ROUGE_L↓ | CIDEr↓ | SPICE↓ |
|---|---|---|---|---|---|---|
| ✓ | × | 0.692 | 0.247 | 0.551 | 1.136 | 0.195 |
| × | ✓ | 0.612 | 0.208 | 0.493 | 0.868 | 0.160 |
| ✓ | ✓ | **0.605** | **0.204** | **0.483** | **0.811** | **0.156** |

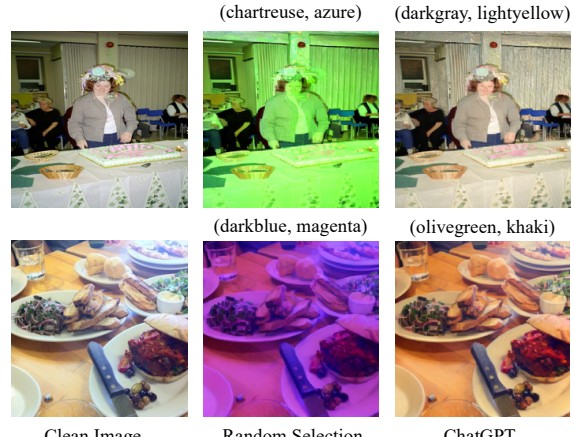

**Figure 7:** Visualization of the generated adversarial examples based on randomly selected and GPT-recommended lighting parameters (start color $c_s$, end color $c_e$).

our model with the model using randomly selected lighting parameters. Table 3 and Fig. 7 provide the quantitative and qualitative results. It is observed that the two models obtain comparable performance, while the generated adversarial relighting image based on GPT has a better visual perception than random selection. Since GPT has powerful reasoning ability, it can recommend lighting colors that are more in line with the visual perception of the clean images. Such results demonstrate the effectiveness of the developed ChatGPT -driven lighting parameter selection.

**Evaluation of Two-Step lighting-based Optimization.** We conduct an ablation test to verify the effectiveness of the proposed two-step lighting-based optimization operations (lighting parameter and lighting image optimization) in Fig. 4. Using parameter optimization independently results in a specific level of attack performance. However, when solely utilizing light image optimization, superior attack performance is attained compared to parameter optimization. Intriguingly, combining parameter and lighting image optimization yields the most impressive performance.

## 6. Conclusion

We propose LightD, a natural GPT-driven relighting attack against VLP models via a pre-trained relighting model. By leveraging the strengths of ChatGPT to generate plausible lighting scenarios and SGA to optimize adversarial effects, LightD achieves impressive results in fooling VLP models. Furthermore, we propose a general optimization framework for adapting existing natural adversarial attacks for image classification to VLP models, experimental results underscore its versatility and applicability across different tasks. Comprehensive comparisons are conducted to verify the effectiveness of the proposed LighD with existing non-suspicious adversarial attacks on various VLP models for image captioning and visual question-answering tasks.

## Impact Statements

This study contributes to the broader field of AI safety by demonstrating the potential vulnerabilities of vision-language pre-training (VLP) models to non-suspicious adversarial attacks. By developing LightD, a GPT-driven adversarial relighting framework, we provide a novel method for assessing and enhancing the robustness of VLP models against real-world threats.

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

## A. Summary of the Appendix

In this appendix, we provide more details of the GPT template, the SGA (Lu et al., 2023) optimization method, experimental setups, optimal parameter selection, quantitative and qualitative comparison results of image captioning tasks on the Flick8K dataset, and more visual results on both image captioning and visual question-answering (VQA) tasks.

## B. GPT Template

We employ GPT to accommodate the initial lighting parameters (e.g., start color, end color, and light direction) to generate the reference lighting image for the relighting procedure of IC-Light (Zhang et al., 2025). The basic motivation lies in GPT will output optimal lighting colors that are consistent with the clean image via carefully analyzing it. The template for GPT is illustrated in Fig. 8.

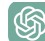

Prompt: Hello GPT…In this image, there are multiple objects, each with its unique color and the background it resides in. Carefully analyze these colors and consider how different lighting conditions could make it difficult to distinguish the objects from one another or from their backgrounds. Your task is to accommodate two colors from the color palette: one as the starting color and the other as the ending color. You may choose darker colors, such as black, to make the image more challenging to interpret. The lighting will gradually transition from the starting color to the ending color, so you also need to accommodate a lighting direction from the lighting direction palette. Ensure that both colors are chosen from the provided color palette and that the lighting direction is selected from the lighting direction table. The combination of these two colors and the specified lighting direction should effectively obscure the details in the image. The output format should be: starting color, ending color, lighting direction (all on one line, separated by commas). Please ensure the output contains no additional or irrelevant content. Approach the task step by step.

**Figure 8:** GPT template for initial lighting parameter selection.

## C. SGA Optimization

In this study, we leverage the core idea of SGA (Lu et al., 2023) to deal with the optimization problem because it can enhance the diversity of adversarial examples along the optimization path by augmenting image-text pairs. During the optimization procedure, let $I_i$ denote the generated adversarial image at the $i$th step. SGA conducts a data augmentation by resizing $I_i$ into multiple resolutions $M$, resulting in $I_i = \{I_{i1}, I_{i2}, \ldots, I_{iM}\}$, the iteration process is defined as

$$I_{i+1} = I_i + \alpha \cdot sign\left(\frac{\nabla_I \sum_{j=1}^{M} \mathbf{J}(f_\phi(I_{ij}), g_\varphi(T))}{\| \nabla_I \sum_{j=1}^{M} \mathbf{J}(f_\phi(I_{ij}), g_\varphi(T))\|}\right) \tag{8}$$

where $\mathbf{J}$ denotes the objection function, $T$ is the label text; $f_\phi$ and $f_\varphi$ represent the image encoder and text encoder of the multimodal model, respectively.

## D. More Details of Experimental Setups

Here, we give more details about setups, including datasets, baselines, victim VLP models, and evaluation metrics.

**Datasets.** In this study, we demonstrate the effectiveness of our techniques for crafting adversarial examples against open-source VLP models on two typical downstream vision-and-language (V+L) tasks: image captioning and VQA. Three widely used multimodal image captioning datasets are leveraged in this study, including MSCOCO (Lin et al., 2014), Flickr8K (Hodosh et al., 2013), and Flickr30K (Plummer et al., 2015). For the VQA task, MSCOCO and DAQUAE (Malinowski & Fritz, 2014) datasets are employed. In this study, we randomly choose 1,000 images from the test set of the above datasets as clean images to craft adversarial examples. The detailed information is listed as follows:

- MSCOCO dataset can be adopted for both image captioning and VQA tasks. MSCOCO encompasses a total of 123,287 images, each image being annotated with approximately five captions according to human engineering, providing prolific linguistic annotations that describe the visual content with different degrees of detail and perspective. Moreover, each image in the MSCOCO dataset has three questions, each question has ten corresponding human-generated answers.

- Flickr8K dataset contains 8,092 images, each accompanied by five descriptions. These descriptions, crafted by human annotators, provide detailed natural language annotations that capture various aspects of the images, including objects, actions, and contextual elements.

- Flickr30K dataset developed as an expanded version of the Flickr8K dataset, it involves 31,783 images, and each image contains five human-written descriptions that capture a wide range of visual details.

- DAQUAR includes around 12,468 questions and answers, with each question paired with a corresponding answer.

**Baseline Methods.** We evaluate the proposed method with the state-of-the-art natural adversarial attack methods: three adversarial relighting attacks and three adversarial color attacks. Adversarial relighting attacks refer to modifying the lightness and brightness of the images, including ALA (Huang et al., 2023), EdgeFool (Shamsabadi et al., 2020a), and Jadena (Gao et al., 2022). To obtain adversarial examples with a high attack success rate, ALA (Huang et al., 2023) proposes unconstrained enhancement in terms of the light and shade relationship in images. To enhance the naturalness of images, ALA crafts the naturalness-aware regularization according to the range and distribution of light. EdgeFool (Shamsabadi et al., 2020a) generates adversarial images with perturbations that enhance image details via training a fully convolutional neural network end-to-end with a multi-task loss function. Jadena (Gao et al., 2022) jointly and locally tunes the exposure and additive perturbations of the image according to a newly designed high-feature-level contrast-sensitive loss function. Adversarial color attacks attempt to change image color to obtain adversarial examples, including SemanticAdv (Hosseini & Poovendran, 2018), ColorFool (Shamsabadi et al., 2020b), and AdvCF (Zhao et al., 2023). SemanticAdv (Hosseini & Poovendran, 2018) crafts adversarial images as a constrained optimization problem and develops an adversarial transformation based on the shape bias property of the human cognitive system. ColorFool (Shamsabadi et al., 2020b) generates unrestricted perturbations by exploiting image semantics to selectively modify colors within chosen ranges that are perceived as natural by humans. AdvCF (Zhao et al., 2023) is a color transformation attack that is optimized with gradient information in the parameter space of a simple color filter.

**Victim VLP Models.** For the image caption task, we employ three typical VLP models to verify their robustness against adversarial attacks, including CLIPCap (Mokady et al., 2021), BLIP (Li et al., 2022), and BLIP2 (Li et al., 2023a) are used. For the VQA task, we use BLIP and BLIP2. Specifically, CLIPCap incorporates a lightweight transformer-based architecture to generate captions from the CLIP embeddings. Unlike traditional image captioning models that rely on training a large neural network from scratch, CLIPCap achieves high-quality captioning performance with a relatively smaller and more efficient model that can generate accurate and contextually rich captions. BLIP is designed to unify several vision-language tasks within one architecture. Unlike models that require separate setups or fine-tuning for each task, BLIP can be adapted seamlessly to multiple tasks without significant architectural changes. This makes it more efficient and flexible, especially for research or applications requiring versatility across visual-language tasks. BLIP-2 is an advanced multimodal model developed to extend the capabilities of the original BLIP model, offering improved efficiency and performance for a wide range of V+L tasks, such as image captioning, VQA, and other open-ended reasoning tasks.

**Evaluation Metrics.** The image captioning task typically utilizes BLEU (Naseer et al., 2021), METEOR (Banerjee & Lavie, 2005), ROUGE (Chin-Yew, 2004), CIDEr (Vedantam et al., 2015), and SPICE (Anderson et al., 2016) to assess the quality and relevance of the generated captions about reference captions. BLEU measures the similarity between two texts based on different lengths of n-grams (i.e., the number of consecutive words). METEOR calculates the semantic similarity and text alignment of each word. ROUGE metrics include recall, precision, and F-measure, which measure the relevance, similarity, and weighted average of similarity. CIDEr calculates the cosine similarity of N-grams and considers Term Frequency-Inverse Document Frequency weights to differentiate the importance of different N-grams. SPICE assesses quality by comparing the matching degree of semantic propositions, such as the presence, attributes, and relationships of objects. For the VQA task, the average prediction accuracy (APA) and WUPS (Kafle & Kanan, 2017) are employed to measure the model's performance. APA measures the percentage of successful prediction answers among all the images. WUPS measures how much a predicted answer differs from the ground truth based on the difference in their semantic meaning. We employ a no-reference image quality index to assess the naturalness of the generated adversarial images, i.e., NIQE (Mittal et al., 2012). A smaller value of the NIQE metric represents a better visual quality of the image.

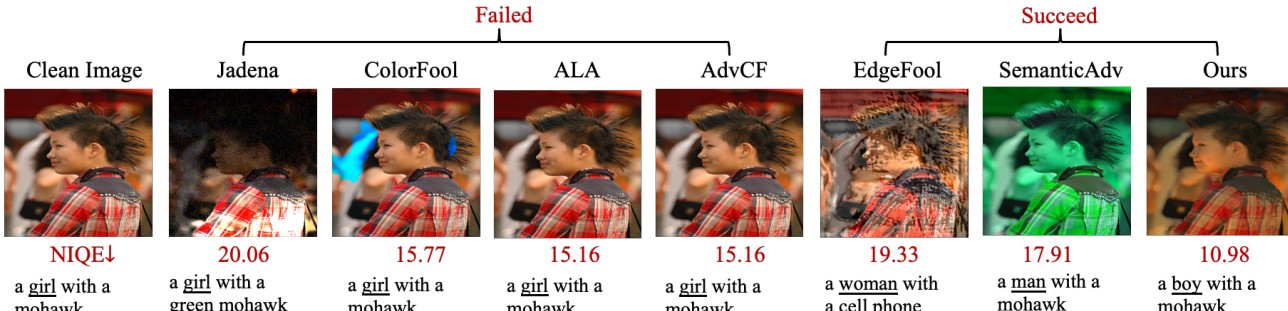

**Figure 9:** Visualization of adversarial examples of attacking BLIP model on Flickr8K in image captioning.

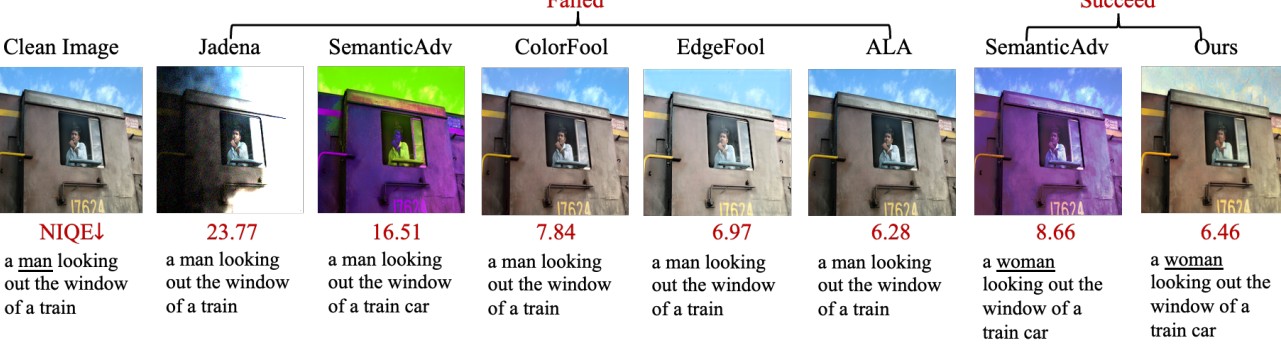

**Figure 10:** Visualization of adversarial examples of attacking BLIP2 model on Flickr30K in image captioning.

## E. Optimal Parameter

**Table 5: Evaluation of resizing number $M$.**

| $M$ | Attack Performance | | | | | Visual Naturalness |
| | BLEU↓ | METEOR↓ | ROUGE$_L$↓ | CIDEr↓ | SPICE↓ | NIQE↓ |
|---|---|---|---|---|---|---|
| 1 | 0.616 | 0.217 | 0.511 | 0.954 | 0.174 | 8.868 |
| 3 | 0.607 | 0.208 | 0.486 | 0.836 | 0.161 | **8.258** |
| 5 | 0.605 | 0.204 | 0.483 | 0.811 | 0.156 | 8.352 |
| 7 | **0.596** | **0.198** | **0.477** | **0.817** | **0.154** | 8.412 |

In the proposed methodology, the parameter $M$, representing the number of resizing iterations for optimizing the reference light image, is adjustable. We conduct targeted experiments to ascertain the optimal parameter setting and assess the influence on the efficacy of our proposed approach. Specifically, we utilize the proposed method to craft adversarial examples to attack the BLIP2 model for image captioning tasks on the MSCOCO dataset. Table 5 presents the outcomes of experiments with varying resizing iterations. This table shows that the attack performance improves as the number of resizing iterations increases. However, more resizing numbers means requiring more computational cost. To balance the proposed method's attack performance, visual naturalness, and computational efficiency, we select the $M = 5$ for this study.

## F. Performance Comparison on Flickr8K

We present the comparison results of all attack methods on the Flickr8K dataset for image captioning tasks.

**Quantitative Comparison.** Table 6 shows all the involved methods in the image captioning task on the Flickr8K dataset. From this table, we can observe that all the existing non-suspicious adversarial attacks for image classification tasks can be transferred to image captioning tasks according to the proposed general optimization strategy. However, these attacks do perform not well in balancing the attack performance and visual naturalness simultaneously. In contrast, the proposed attack achieves the best adversarial attack performance while retaining visual naturalness on all VLP models on the Flickr8K dataset. It demonstrates the effectiveness and superiority of the proposed model based on the pre-trained relighting

Table 6: Comparison with state-of-the-art methods for image captioning task on Flickr8k dataset.

| Model | Attack | BLEU | METEOR | ROUGE$_L$ | CIDEr | SPICE | NIQE |
|---|---|---|---|---|---|---|---|
| **CLIPCap** | SemanticAdv | 0.465 | 0.147 | 0.367 | 0.198 | 0.080 | 9.228 |
| | ColorFool | 0.507 | 0.160 | 0.389 | 0.256 | 0.091 | 9.363 |
| | AdvCF | 0.447 | 0.133 | 0.344 | 0.170 | 0.070 | 9.596 |
| | EdgeFool | 0.441 | 0.120 | 0.333 | 0.143 | 0.056 | 16.707 |
| | ALA | 0.558 | 0.178 | 0.416 | 0.374 | 0.116 | 9.707 |
| | Jadena | 0.524 | 0.152 | 0.383 | 0.272 | 0.090 | 19.255 |
| | **LightD(Ours)** | **0.401** | **0.095** | **0.300** | **0.088** | **0.039** | **8.240** |
| **BLIP** | SemanticAdv | 0.568 | 0.194 | 0.428 | 0.450 | 0.127 | **5.510** |
| | ColorFool | 0.588 | 0.202 | 0.444 | 0.501 | 0.136 | 5.634 |
| | AdvCF | 0.571 | 0.195 | 0.433 | 0.445 | 0.129 | 5.608 |
| | EdgeFool | 0.515 | 0.156 | 0.383 | 0.313 | 0.097 | 7.626 |
| | ALA | 0.669 | 0.241 | 0.505 | 0.753 | 0.175 | 5.664 |
| | Jadena | 0.611 | 0.207 | 0.449 | 0.563 | 0.140 | 9.521 |
| | **LightD(Ours)** | **0.463** | **0.132** | **0.353** | **0.193** | **0.074** | 5.926 |
| **BLIP2** | SemanticAdv | 0.619 | 0.203 | 0.478 | 0.557 | 0.142 | 9.523 |
| | ColorFool | 0.652 | 0.211 | 0.495 | 0.623 | 0.153 | 9.621 |
| | AdvCF | 0.614 | 0.198 | 0.476 | 0.542 | 0.141 | 9.952 |
| | EdgeFool | 0.668 | 0.224 | 0.522 | 0.723 | 0.161 | 12.122 |
| | ALA | 0.690 | 0.231 | 0.530 | 0.767 | 0.171 | 9.752 |
| | jadena | 0.644 | 0.201 | 0.486 | 0.606 | 0.140 | 19.251 |
| | **LightD(Ours)** | 0.599 | **0.178** | **0.454** | **0.503** | **0.128** | **8.298** |

model. Moreover, the specifically designed GPT-driven lighting parameter selection and SGA-based two-step collaboration optimization enable craft natural and non-suspicious adversarial relighted images with promising attack capability.

**Qualitative Comparison.** To illustrate the advantage of our method in detail, we visualize the adversarial examples and predicted captions of all the involved attack methods on the Flick8K dataset in Fig. 9. The adversarial examples are generated by attacking the ClIPCap model. Upon observing the compared non-suspicious adversarial attacks, it is evident that they are unable to attain both optimal attack performance and visual naturalness concurrently. Specifically, while some attacks achieve promising visual quality, they fail to deceive the VLP models. Conversely, other attacks may successfully compromise the models, but the resulting adversarial images suffer from poor visual quality and can be easily detected by human observers. On the contrary, our method excels in achieving high attack performance while maintaining visual naturalness in attacking both CLIPCap and BLIP2 models for image captioning tasks.

## G. More Visualizations on V+L Tasks

**Image Captioning.** we provide the visual results of all methods in attacking the BLIP2 model on Flickr30K datasets in Fig. 10. The generated adversarial relighted images of the proposed method enable the ability of misleading BLIP models for image captioning while maintaining the visual quality and naturalness on Flickr30K datasets.

**VQA.** We present more visual results of the non-suspicious adversarial attacks for VQA. Fig. 11 and Fig. 12 show the adversarial examples of attacking the BLIP and BLIP2 model on the DAQUAR dataset, respectively. The visual results verify the effectiveness and superiority of our method in terms of both attack performance and visual naturalness.

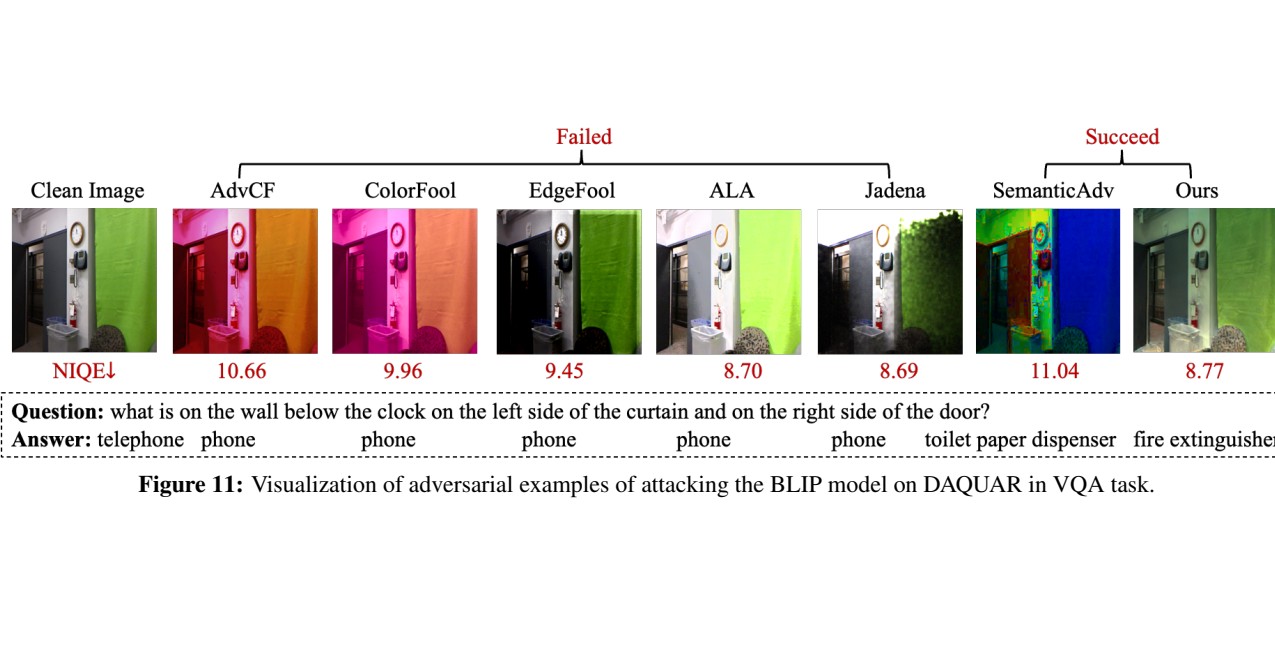

**Figure 11:** Visualization of adversarial examples of attacking the BLIP model on DAQUAR in VQA task.

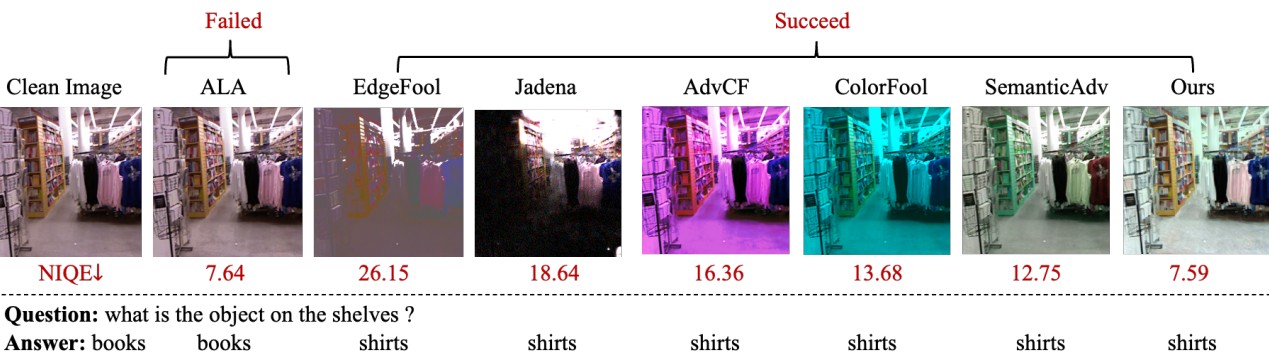

**Figure 12:** Visualization of adversarial examples of attacking the BLIP2 model on DAQUAR in VQA task.

