# OpenReview forum: "Light as Deception: GPT-driven Natural Relighting Against Vision-Language Pre-training Models"
_ICML.cc/2025/Conference — Submitted to ICML 2025_

### Official Review · Reviewer_j7H2 · 2025-03-10

**Overall Recommendation:** 2

**Summary:**

In this work, the authors propose a GPT-driven adversarial relighting framework called LightD to deceive VLP models by adjusting the lighting of clean images. LightD first adopts ChatGPT to select the lighting parameter, then optimizes the parameters of relighting model IC-Light to relight the input image.

**Claims And Evidence:**

yes

**Essential References Not Discussed:**

N/A

**Experimental Designs Or Analyses:**

Yes. The results on MSCOCO validate the effectiveness of LightD and the ablation studies support the importance of ChatGPT.

**Methods And Evaluation Criteria:**

yes

**Other Comments Or Suggestions:**

N/A

**Other Strengths And Weaknesses:**

1. What is the difference between the color attack of image classification and VLP models? The authors claim that it is the first work to investigate such an attack against VLP models. However, I do not see significant difference here. And there are many similar works for image classification.

2. The motivation is not clear. In my opinion, only the initialization is novel for me. The relighting process is just a PGD attack process.

2. I am curious why ChatGPT helps. It does not access the victim VLP models. Is it the best way to conduct such an attack, or does it just outperform random initialization by a small margin?

3. It would be great if the authors could also show the effectiveness of LightD on image classification tasks, such as CNNs and ViT, to further show the generality of LIghtD.

**Questions For Authors:**

N/A

**Relation To Broader Scientific Literature:**

Adopting ChatGPT to select the starting parameters is interesting.

**Theoretical Claims:**

N/A

---

> ### Author Rebuttal · Authors · 2025-03-31
>
> ### 1. Difference between color attack classification vs. VLP
> Thank you for the important question. Attacking VLP models differs significantly from attacking classifiers due to the following:
>
> **a）Cross-modal structure.** VLPs generate captions or answers, requiring joint reasoning over vision and language. Attacks must target this multimodal alignment, not just label prediction.
>
> **b) semantic alignment instead of label prediction.**
> VLP models are trained to align visual features with textual descriptions. Simply changing the image color may not disrupt this alignment, as the model can rely on robust, multimodal embeddings.
>
> **c) Increased Robustness.** Pretrained VLPs like BLIP2 or CLIP are trained with large-scale vision-text pairs and are often more robust to simple perturbations. As we show in our experiments, existing color-based attacks (e.g., ColorFool, AdvCF) achieve limited effectiveness when directly applied to VLP tasks like captioning or VQA.
>
> We address these challenges by:
>
> **a) Cross-modal attack objective.** We design a cross-modal attack objective that specifically targets the semantic misalignment between relighted images and their textual outputs, something not addressed by classification-based color attacks.
>
> **b) GPT-based reference relighting.** We adopt a reference relighting approach based on GPT accommodation, enabling more flexibility in generating semantically disruptive but visually coherent transformations.
>
> **c) Two-step collaborative optimization.** It effectively expands the capabilities of our model by first optimizing lighting parameters and then refining the reference lighting image, achieving robust adversarial effects while preserving visual naturalness.
>
> Generalization to classification tasks:
> We evaluate LightD on CLIP with ImageNet to test its applicability to the classification task. Compared with ALA, AdvCF, and ColorFool, LightD achieves significantly lower Top-1/Top-5 accuracy while maintaining better perceptual quality (NIQE):
>
> |Model|Top1_acc|Top5_acc|NIQE|
> |-|-|-|-|
> |Clean Image|0.601| 0.842|9.530|
> |ColorFool| 0.183| 0.615|9.884|
> |AdvCF| 0.063|0.404|10.068|
> |ALA| 0.565| 0.838|9.824|
> |**LightD(Ours)**|**0.043**|**0.107**|**8.429**|
>
> ### 2. Contribution
> We respectfully disagree with the notion that our method is simply a "PGD attack with a novel initialization." While our optimization process shares the conceptual structure of gradient-based attacks, w would like to emphasize the contributions of our work from the following perspectives:
>
> **a) Novel problem setting.** To the best of our knowledge, we are the first to explore non-suspicious adversarial attacks via natural relighting against VLPs. Previous works mainly target classification tasks using $L_p$-bounded noise.
>
> **b) Flexible framework.** Our pipeline is modular and extensible; IC-Light and SGA can be swapped for other relighting or optimization techniques. The ChatGPT-guided initialization enhances semantic alignment, while the two-step optimization bridges visual quality and attack effectiveness.
>
> **c) Cross-modal integration.** Though individual components may appear simple, combining them for cross-modal attacks (e.g., on captioning or VQA) is non-trivial. Our joint loss formulation (Eq. 7) enables this by balancing visual coherence with semantic misalignment.
>
> ### 3. Impact of ChatGPT
> Thank you for your comment. While ChatGPT does not directly interact with the victim VLP models, we find that its role as a semantic prior generator is highly beneficial to the attack pipeline.
>
> As shown in Table 3, lighting parameter initialization via ChatGPT consistently yields higher attack effectiveness across multiple VLP models compared to random initialization. In Figure 7, we further illustrate a qualitative example where the ChatGPT-based suggestion results in a more natural and semantically coherent relighting effect. These improvements stem from ChatGPT's strong understanding of scene structure, lighting conventions, and human visual perception.

---

### Official Review · Reviewer_bMde · 2025-03-14

**Overall Recommendation:** 4

**Summary:**

This paper introduces LightD, a novel GPT-driven adversarial relighting technique against VLP models. It uses ChatGPT to generate lighting scenarios and uses SGA to optimize adversarial effects. They also propose a general optimization framework for adapting existing natural adversarial attacks for image classification to VLP models with extensive experimental results showing its validity.

**Claims And Evidence:**

The key claims are supported by clear evidences.

**Essential References Not Discussed:**

Most of the key references are discussed.

**Experimental Designs Or Analyses:**

The experimental design and analysis are sound.

**Methods And Evaluation Criteria:**

The methods and evaluation criteria make sense.

**Other Comments Or Suggestions:**

N/A

**Other Strengths And Weaknesses:**

Strength:
1. Novelty: the paper is the first work to investigate the effectiveness of non-suspicious adversarial attacks against VLP models.
2. The experiments are quite extensive with good results.
3. The adversarial images generated by LightD are natural visually and can achieve attack effectiveness.

Weaknesses:
1. LightD shows higher NIQE score for the BLIP model consistently (sometimes get the worst results). There is no explanation for why this could happen.

2. Would the two-step optimization process induces additional computation cost? How expensive is it compared to other baseline methods.

**Questions For Authors:**

See weaknesses.

**Relation To Broader Scientific Literature:**

The work connects the relighting attacks from image classification tasks to VLP model.

**Theoretical Claims:**

The paper does not make any theoretical claims.

---

> ### Author Rebuttal · Authors · 2025-03-31
>
> ### 1. NIQE result analysis on BLIP model
> Thank you for pointing this out. NIQE is a widely adopted no-reference metric in the field and is also used by the baselines for fair comparison. However, it is not perceptually linear and may not be highly sensitive to subtle semantic-preserving perturbations. In some adversarial settings, small differences in NIQE scores can sometimes correspond to more noticeable perceptual variations or vice versa. Therefore, we also provide qualitative results to complete the quantitative results in the paper.
>
> In certain VLP+dataset combinations (such as Flickr30K + BLIP), the captioning model’s semantic sensitivity may prompt our optimization strategy toward more aggressive relighting variations to ensure successful attacks. These changes, while still visually acceptable, may deviate from the "natural statistics" that NIQE favors, leading to relatively higher NIQE values.
>
> ### 2. Computation cost
> Thanks for this comment. The following table shows the computation cost of all involved models. LightD relies on IC-Light, a pre-trained diffusion-based relighting model, to generate semantically consistent and physically plausible lighting variations. This naturally introduces additional inference time due to the sampling process inherent in diffusion models. The additional computation enables greater visual naturalness, semantic alignment, and stronger attack effectiveness, which are core goals of non-suspicious adversarial attacks.
>
> | Model     | SemanticAdv | ColorFool | AdvCF | EdgeFool | ALA   | Jadena | LightD (Ours) |
> |-----------|-------------|-----------|-------|----------|-------|--------|---------------|
> | images/sec| 12.79       | 7.73      | 8.46  | 6.61     | 10.32 | 1.77   | 35.21         |

---

> > ### Comment · Reviewer_bMde · 2025-04-03
> >
> > Thank you for the response which addresses my concern. I will maintain my positive score.

---

> > > ### Author Response · Authors · 2025-04-05
> > >
> > > Appreciate it! We respectfully thanks for your support and your decision to maintain a positive attitude towards our work.

---

### Official Review · Reviewer_3ss3 · 2025-03-16

**Overall Recommendation:** 2

**Summary:**

This paper proposed a framework for generating natural adversarial samples for VLP models via semantically guided relighting. LightD leverages ChatGPT for context-aware lighting parameters and integrates a pretrained relighting model (IC-light). A gradient-based refinement further enhances adversarial impact while preserving visual realism.

**Claims And Evidence:**

Overall, the claims presented in the submission are clear.

**Essential References Not Discussed:**

N/A

**Experimental Designs Or Analyses:**

The submission lacks comparisons with recent color-based attack methods. The chosen baselines are somewhat outdated, as all were introduced more than two years ago.

While the authors cite ALA and AdvCF as being released in 2023, these methods were actually first proposed on ArXiv in 2022 and 2020, respectively.

Additionally, as an attack method, the paper does not include experiments on transferability or robustness against defense techniques. These aspects are crucial in evaluating the effectiveness of modern attack methods and should be considered to provide a more comprehensive assessment.

**Methods And Evaluation Criteria:**

Overall, the proposed method appears reasonable for a feasible color-based attack.

**Other Comments Or Suggestions:**

Please refer to the above comments.

**Other Strengths And Weaknesses:**

### **Strengths:**
1. The writing is clear, making the ideas easy to follow.
2. The proposed method is reasonable and feasible, demonstrating performance benefits for the attack.
3. Experiments indicate that the proposed method achieves advantages over some existing approaches in certain aspects.

### **Weaknesses:**
1. **Limited contribution to the field:** The work primarily integrates IC-Light into the attack generation pipeline without substantial novelty. It does not offer significant new insights beyond prior works that optimize generator parameters or image attributes, resulting in only marginal benefits to the adversarial attack field.
2. **Unconvincing baseline comparisons:** The chosen comparison methods are relatively outdated, making it difficult to accurately position the proposed approach within the field. More recent baselines would strengthen the evaluation.
3. **Unsubstantiated claim in the abstract:** The authors mention that existing works have "restricted optimization spaces", but the paper lacks a clear explanation or supporting analysis to justify this claim.
4. **Absence of a limitations discussion:** The proposed approach heavily depends on the performance of IC-Light and ChatGPT. A dedicated analysis of the method’s stability and potential limitations would provide a more balanced perspective.
5. **Lack of transferability and robustness evaluation:** Modern adversarial attack methods are typically assessed based on their transferability and robustness against defense techniques. This paper does not explore these crucial aspects, limiting the comprehensiveness of its evaluation.

**Questions For Authors:**

Please refer to the above comments.

**Relation To Broader Scientific Literature:**

From my perspective, this work does not present a significant contribution. It primarily integrates the recent IC-Light method into the color-based adversarial attack domain in a straightforward manner, without substantial innovation or novel insights.

**Theoretical Claims:**

Not applicable. There is no theoretical contribution in the paper.

---

> ### Author Rebuttal · Authors · 2025-03-31
>
> ### 1. Contribution
> We respectfully disagree with the assessment that our work lacks significant contribution. While our approach does utilize IC-Light, we would like to clarify several important points. First, scientific advancement often comes from novel combinations and applications of existing technologies rather than entirely new tools. Our contribution lies not in creating a new relighting model, but in addressing the previously unexplored challenge of leveraging relighting techniques for natural adversarial attacks against VLP models. Second, our integration is far from straightforward. We encountered and solved several significant challenges:
>
> **a) Semantic preservation.** Default IC-Light outputs (e.g., text-/background-conditioned) often alter semantic content. We propose a reference-based relighting approach that preserves semantics by introducing only lighting differences.
>
> **b) Cross-modal adversarial optimization.** Achieving attacks that mislead VLPs while maintaining naturalness required a carefully designed optimization objection.
>
> **c) Contextual parameter selection.** We are the first to use LLM for semantically aligned lighting parameter initialization for adversarial relighting, allowing coherent image-specific manipulations.
>
> We also emphasize the contributions of our work from the following perspectives:
>
> **a) Novel problem setting.** To the best of our knowledge, we are the first to explore non-suspicious adversarial attacks via natural relighting against VLPs. Previous works mainly target classification tasks using $L_p$-bounded noise.
>
> **b) Flexible framework.** Our pipeline is modular and extensible; IC-Light and SGA can be swapped for other relighting or optimization techniques. The ChatGPT-guided initialization enhances semantic alignment, while the two-step optimization bridges visual quality and attack effectiveness.
>
> **c) Cross-modal integration.** Though individual components may appear simple, combining them for cross-modal attacks (e.g., on captioning or VQA) is non-trivial. Our joint loss formulation (Eq. 7) enables this by balancing visual coherence with semantic misalignment.
>
> ### 2. Compared methods
> Thank you for the suggestion. To the best of our knowledge, the color-based adversarial attack methods included in our evaluation are still among the most widely cited and representative non-suspicious color transformation attacks in the literature. We truly welcome suggestions for more recent or under-reviewed methods that we may have missed. We are happy to incorporate additional baselines in the final version or future iterations of this work to ensure a fair and up-to-date evaluation.
>
> ### 3. Transferability and robustness against defense
> **a) Cross-model transferability.**
> We evaluate the transferability of LightD-generated adversarial examples on MSCOCO for image captioning and VQA, where adversarial examples were generated on one VLP model and tested on a different model. Results (https://imgur.com/a/NG2l77b) show significant performance drops across unseen models, confirming that our attack generalizes across architectures.
>
> **b) Defense robustness.**
> We tested LightD against common input transformation defenses (grayscale conversion to remove color information and median filtering for denoising). Results (https://imgur.com/a/LH1r8I2) show that while these reduce attack strength to some extent, LightD remains substantially more effective than baselines, highlighting its robustness under simple defenses.
>
> We will provide these transferability and robustness against defense experiments in the final version.
>
> ### 4. Optimization spaces explanation
> Existing non-suspicious attacks typically rely on low-dimensional global transformations (e.g., hue, brightness), which limit perturbation diversity and attack power.
> In contrast, LightD introduces a two-stage optimization: 1) Lighting parameters, 2) Pixel-level adjustment of the reference lighting image. This extends the optimization from a small parameter set to an entire image matrix, greatly expanding the perturbation space while preserving realism. As shown in our ablation study (Table 4), pixel-level refinement significantly improves performance without degrading visual quality—supporting the importance of a richer optimization space.
>
> ### 5. Limitations discussion
> Thank you for the thoughtful suggestion. We will include a Limitations section in the final version. We acknowledge two key limitations:
> **a) Reliance on IC-Light.**
> Our results depend on IC-Light's relighting quality. However, the framework is modular and can adopt alternative models, which we plan to explore in future work.
> **b) Dependence on ChatGPT.**
> ChatGPT provides semantically meaningful initialization but introduces some response variability. We mitigate this using fixed templates and normalization, but future work may involve a dedicated learnable lighting predictor for more stable, task-specific initialization.

---

> > ### Comment · Reviewer_3ss3 · 2025-04-05
> >
> > Thanks for the authors' responses. The response has alleviated some of my earlier concerns regarding the experimental integrity, particularly in terms of transferability and robustness to defenses. However, several concerns still remain.
> >
> > My primary concern is still with the contribution of the paper. Based on the authors’ response and their discussions with other reviewers, I understand that the authors aim to position their work as “the first non-suspicious adversarial attack via natural relighting against VLPs”. While this may hold under a narrow definition, the technical novelty appears limited. The method primarily shifts the optimization target from conventional image parameters or generator parameters to those of a more powerful relighting model. This substitution, while effective, feels straightforward and lacks deeper methodological insight. Consequently, I find the level of innovation and the conceptual contribution of the work to be insufficient.
> >
> > Regarding related work, especially recent color-based adversarial attacks, I would like to highlight that the authors should consider including more up-to-date comparisons. For instance, Natural Color Fool (NCF) [1] and other works published post-2022 are more relevant than those from 2020 or earlier.
> >
> > As for the limitations discussion, it would be more convincing to support claims with experimental evidence rather than a brief textual explanation. For example, in the discussion regarding the use of ChatGPT, it is important to clarify how the non-determinism of ChatGPT’s outputs affects the consistency or success of the attack. An empirical analysis of such variance would strengthen the robustness evaluation of the method.
> >
> > [1] Yuan, Shengming, et al. "Natural color fool: Towards boosting black-box unrestricted attacks". Advances in Neural Information Processing Systems 35 (2022): 7546–7560.

---

> > > ### Author Response · Authors · 2025-04-07
> > >
> > > Thank you again for your thoughtful feedback. We would like to provide additional clarification regarding the novelty and contribution of our work, suggested baseline, and our analysis of the limitations.
> > >
> > > >***Our motivation, research principle, and contribution***
> > >
> > > We believe that **innovation in research is not only about proposing entirely new algorithms or theories**. A large part of scientific progress comes from designing general, effective frameworks that combine modern tools in meaningful ways. **Our research philosophy is simple but effective**. In this work, we present a general and flexible adversarial attack framework that simulates how a human attacker might naturally reason by subtly and harmoniously adjusting lighting in a way that triggers semantic confusion while preserving the visual realism of the image.
> > >
> > > The core idea is straightforward yet powerful. Vision-language pretraining (VLP) models are now central to many vision and cross-modal tasks, **not a narrow or specialized research area**. Despite this, most existing adversarial attack methods either rely on pixel-level perturbations or assume white-box settings, and are typically designed for classification. Our framework is the first to explicitly target VLPs using relighting-based, non-suspicious attacks.
> > >
> > > We chose IC-Light not arbitrarily but because it is a state-of-the-art relighting model (the top score in ICLR 2025), capable of precise, semantics-preserving lighting edits. Its strong control over natural illumination made it a natural candidate for adversarial use. We believe good research builds on powerful tools in creative ways, namely **standing on the shoulders of giants to tackle new problems**. To our knowledge, our work is the first to realize this idea concretely in the adversarial context.
> > >
> > >
> > > We found that lighting manipulation alone is insufficient without semantic guidance. As vision-language models grow more capable and general, we leverage GPT-4o to suggest initial lighting parameters, drawing on its strong visual commonsense to align lighting with image content. With the emergence of powerful, affordable multimodal models like DeepSeek-VL, this approach is increasingly practical. Our framework harnesses this trend, paving the way for a new class of accessible, effective, and semantically grounded attacks.
> > >
> > > In addition, we implemented a general pipeline to adapt several natural attack baselines originally designed for image classifiers. This engineering effort establishes, for the first time, a benchmark for evaluating natural adversarial attacks on VLP models. We hope this contributes to a useful resource for the community and encourages further research on multimodal robustness under realistic perturbations.
> > >
> > > ---
> > >
> > > >***Comparison with NCF***
> > >
> > > For the baseline you suggested (NCF, 2022), we extend it to attacking VLP, the result in image captioning and VQA tasks is shown below:
> > >
> > > A. Image captioning (CLIPCap model on MSCOCO dataset)
> > >
> > > | **Attack**       | **BLEU↓** | **METEOR↓** | **ROUGE↓** | **CIDEr↓** | **SPICE↓** |
> > > |------------------|-----------|-------------|------------|------------|------------|
> > > | Clean Image      | 0.738     | 0.267       | 0.539      | 1.094      | 0.198      |
> > > | NCF    | 0.710       | 0.243      | 0.537      | 1.126      | 0.195    |
> > > | **LightD (Ours)**| **0.460** | **0.119**   | **0.340**  | **0.211**  | **0.055**  |
> > >
> > > B. VQA (BLIP2 on MSCOCO)
> > > | **Attack**       | **APA↓** | **WUPS0.9↓** |
> > > |------------------|-----------|-------------|
> > > | Clean Image      | 56.75     | 0.590       |
> > > | NCF    | 48.75       | 0.528      |
> > > | **LightD (Ours)**| **40.31** | **0.453**   |
> > >
> > > These results show that **LightD consistently outperforms NCF across metrics**.
> > >
> > > In addition, some visual examples are shown in https://imgur.com/a/SmFYyWv.
> > >
> > > We will follow your suggestion to add this baseline in the final version.
> > >
> > > ---
> > >
> > > >***The influence of ChatGPT***
> > >
> > > To evaluate ChatGPT's impact, we repeated our experiments five times. The mean values and standard deviations of the model’s performance across different metrics are summarized in the table below. Although minor fluctuations are observed due to ChatGPT's inherent randomness, the results demonstrate that the overall performance remains stable without significant deviations.
> > >
> > > | Times | BLEU  | METEOR | ROUGE_L | CIDEr | SPICE |
> > > |-------|-------|--------|---------|-------|--------|
> > > | 1     | 0.460 | 0.119  | 0.340   | 0.211 | 0.055  |
> > > | 2     | 0.451 | 0.113  | 0.331   | 0.204 | 0.052  |
> > > | 3     | 0.468 | 0.129  | 0.352   | 0.220 | 0.060  |
> > > | 4     | 0.463 | 0.123  | 0.339   | 0.209 | 0.054  |
> > > | 5     | 0.457 | 0.117  | 0.338   | 0.187 | 0.060  |
> > > | **mean** | **0.4585** | **0.118** | **0.339** | **0.199** | **0.0575** |
> > > | **std**  | **4.5E-06** | **2E-06** | **0.000002** | **0.000288** | **0.0000125** |
> > > ---
> > > We respectfully hope you will reconsider your evaluation in light of these clarifications.

---

### Official Review · Reviewer_wa33 · 2025-03-16

**Overall Recommendation:** 3

**Summary:**

The submission proposes a novel, state-of-the-art relighting adversarial attack (LightD) on VLP models using a 4 stage approach:

1) prompt ChatGPT to provide a lighting in the form of lighting parameters (start color, end color, light direction) that could confuse the objects in a given input image

2) generate a reference lighting image using these lighting parameters

3) generate a relit image by providing the input image and reference lighting image to a pretrained relighting model (IC-Light) and optimize the lighting parameters and reference lighting image to improve the adversarial relit image

4) attack the VLP model using the adversarial relit image for tasks such as VQA and image captioning

The proposed technical contributions involving the general framework include two-step SGA-based optimization that first optimizes the lighting parameters and subsequently optimizes the reference lighting image with the objective of maximizing the effectiveness of the adversarial relighting attack. LightD exhibits SoTA performance in adversarial attack effectiveness for vision-language tasks such as VQA and image captioning with reasonable visual naturalness in most settings.

**Claims And Evidence:**

As the claimed contributions in the introduction suggest a general adversarial relighting pipeline with fairly straightforward technical contributions, there are no major issues with the claims in the paper.

**Essential References Not Discussed:**

The paper is reasonably comprehensive citation wise but perhaps there could be some discussion on justifying why IC-Light is the best option for adversarial relighting attacks over other relighting methods could be included in the appendices:

1) Would other relighting methods be worse than IC-Light for adversarial attacks on VLP models or is it not possible to set up a framework using these methods?
- LightIt: Illumination Modeling and Control for Diffusion Models

2) Some relighting methods are only meant for humans and/or faces:
- DifFRelight: Diffusion-based Facial Performance Relighting
- COMPOSE: Comprehensive Portrait Shadow Editing
- Total Relighting
- Face Relighting with Geometrically Consistent Shadows
- Single Image Portrait Relighting
- Deep Single-Image Portrait Relighting
- Towards High Fidelity Face Relighting with Realistic Shadows
- DiFaReli: Diffusion Face Relighting

**Ethical Review Concerns:**

This paper is more focused on introducing a vulnerability of VLP models w.r.t. adversarial relighting attacks and does not focus on offering a solution to the problem. While this may be helpful for alerting or introducing this vulnerability to the community, there was no effort to offer a solution to prevent the attack.

**Ethical Review Flag:**

Flag this paper for an ethics review.

**Ethics Expertise Needed:**

["Inappropriate Potential Applications & Impact  (e.g., human rights concerns)", "Responsible Research Practice (e.g., IRB, documentation, research ethics, participant consent)"]

**Experimental Designs Or Analyses:**

The experiments seem to be thorough and comprehensive: performance is measured across two datasets using 3 VLP models and comparing against 6 strong baselines. Metrics appropriately include measures for adversarial attack effectiveness and visual naturalness. The analysis of experimental results, however, could be more thorough. In particular, there is no discussion for why the NIQE score is worse , or sometimes even the worst, for some combinations of VLP+dataset compared to the baselines.

**Methods And Evaluation Criteria:**

The proposed method is a fairly straightforward approach to the adversarial relighting attack problem that tries to balance a challenging objective: maximizing attack effectiveness on VLP models while preserving visual naturalness. Using existing tools for generating reference lightings, their simple configuration of lighting parameters is a practical decision but nonetheless is limited by the simplicity of the available ComfyUI lighting model. In particular, the reference lighting can only model changes in light color and cannot model other aspects such as light intensity and light size. This will naturally limit the scope of adversarial relit images that can be generated compared to more complicated lighting models or the ability to use environment maps to represent the lighting, potentially compromising both attack effectiveness and visual naturalness.

The proposed evaluation criteria are quite reasonable as they consider both attack effectiveness and visual naturalness and are well-known metrics.

**Other Comments Or Suggestions:**

List of typos in the paper:

1. LightD misspelled in several locations (e.g. in Conclusion as "LighD")
2. "optimization objection" should be "optimization objective" (introduction, line 43)

**Other Strengths And Weaknesses:**

Strengths:

- SoTA performance on adversarial attack effectiveness across multiple datasets and using several SoTA VLP models, baselines compared against are comprehensive (Table 1)

Weaknesses:

- Visual naturalness is inconsistent across experimental settings: NIQE score is not SoTA for some VLP models/dataset combinations and in some cases is the worst among all listed methods (e.g. Flickr30K+BLIP)
- Technical contributions are simple and not particularly insightful: the attack has a straightforward optimization objective and the framework is simple both in terms of lighting model and design. SGA-based optimization is inspired by an existing work, initialization from ChatGPT for reference lighting is a straightforward approach.

**Questions For Authors:**

1. Although the proposed method achieves SoTA performance across most settings, can there be some discussion explaining why the visual naturalness reflected by the NIQE score is worse, or sometimes even the worst, for some combinations of VLP and dataset (as seen in Table 1)?

2. While a novel framework is proposed, the technical contributions and insights within this proposed framework appear to be simple or limited. In particular, SGA-based optimization is inspired by another paper and the remainder of the pipeline is mostly simple ChatGPT-based initialization and a simple lighting parameterization. Can the authors explain and justify how substantial their technical contributions are?

3. There is no discussion about whether there are potentially better design choices (e.g. a more sophisticated lighting parameterization could be used).

**Relation To Broader Scientific Literature:**

The paper could be relevant to both the adversarial attack, deepfake detection, and vision-language domains:

1) It proposes a method that achieves reasonably high visual naturalness in most settings while also achieving SoTA adversarial attack effectiveness on VLP models

2) Deepfake detection methods could use the generated adversarial relit images produced by this method to improve deepfake detection

3) Vision-language models could be trained with this additional adversarial vulnerability in mind

**Theoretical Claims:**

There are no major theoretical claims or proofs in this submission.

---

> ### Author Rebuttal · Authors · 2025-03-31
>
> ### 1. Impact of lighting model
> Thank you for your comment. While existing tools help generate reference lighting, directly applying them in adversarial relighting for VLPs presents challenges:
>
> **1) Optimization vs. semantic integrity.** Complex lighting models can introduce artifacts or distortions, compromising the "non-suspicious" quality needed for effective attacks.
>
> **2) Naturalness vs. attack success.** Simpler lighting manipulations, when applied strategically, can yield strong adversarial effects while maintaining visual realism.
>
> Our LightD framework addresses this via:
>
> **1) ChatGPT-guided parameter selection.** We employ ChatGPT to recommend contextually appropriate lighting parameters that align with image content, enabling targeted manipulation despite the simpler lighting model.
>
> **2) Two-step collaborative optimization.** It effectively expands the capabilities of the simple lighting model by optimizing lighting parameters and reference lighting images, achieving robust adversarial effects while preserving visual naturalness.
>
> We agree that complex models may expand possibilities. Future work includes: 1) Exploring environment maps and richer lighting models. 2) Studying intensity and shape of light sources. 3) Improving optimization for high-dimensional parameter spaces.
>
> ### 2. NIQE result analysis
> Thanks for the comment. NIQE is a non-reference metric, though widely used and adopted by our baselines, is not perceptually linear, and may miss subtle, semantic-preserving changes. Small NIQE differences may not reflect actual perceptual variation. Therefore, we provide qualitative results (e.g., Figs. 5 and 7) to complete the quantitative results.
>
> In some VLP + dataset settings, the semantic sensitivity of captioning models may drive the optimization to adopt stronger relighting changes to ensure attack success. While these are visually acceptable, they may depart from the “natural statistics” that NIQE favors, resulting in higher NIQE scores.
>
> We will include a detailed NIQE analysis in the final version. For future work, we plan to explore alternative perceptual metrics, including learned models and human-in-the-loop evaluations.
>
> ### 3. Performance on other relighting models
> Thank you for the suggestion. Our use of IC-Light over other methods is based on:
>
> **a) Generalizability.** Unlike many relighting models focused on faces or portraits, IC-Light supports diverse categories in our datasets (e.g., animals, scenes, vehicles).
>
> **b) Compatibility with optimization.** IC-Light allows continuous control of lighting parameters (e.g., direction, color gradient), enabling both parameter-level and pixel-level optimization, which is crucial for balancing attack effectiveness and visual naturalness.
>
> **c) Future potential.** We appreciate the mention of LightIt, which introduces promising advances in controllable illumination for diffusion models. Although we currently rely on IC-Light, we plan to explore LightIt once its code and models are available.
>
> We will discuss the applicability of the other relighting methods you mentioned within our proposed framework in the final version.
>
> ### 4. Contribution
> Thank you for recognizing our framework design. While some components (e.g., SGA-based optimization) build on prior work, our key contributions are:
>
> **a) Novel problem setting.** To the best of our knowledge, we are the first to explore non-suspicious adversarial attacks via natural relighting against VLPs. Previous works mainly target classification tasks using $L_p$-bounded noise.
>
> **b) Flexible framework.** Our pipeline is modular and extensible; IC-Light and SGA can be swapped for other relighting or optimization techniques. The ChatGPT-guided initialization enhances semantic alignment, while the two-step optimization bridges visual quality and attack effectiveness.
>
> **c) Cross-modal integration.** Though individual components may appear simple, combining them for cross-modal attacks (e.g., on captioning or VQA) is non-trivial. Our joint loss formulation (Eq. 7) enables this by balancing visual coherence with semantic misalignment.
>
> ### 5. Lighting parameterization design
> Appreciate the comment. While IC-Light supports per-pixel lighting control, we chose a simplified setup (start color, end color, light direction) for the following reasons:
>
> **a) Naturalness and interpretability.** Gradient lighting mimics natural illumination and allows intuitive, controllable manipulation, ensuring coherence and effectiveness.
>
> **b) Avoiding artifacts.** Per-pixel control expands the optimization space and risks introducing unnatural patterns or artifacts, violating the “non-suspicious” constraint. Our approach strikes a balance between expressiveness and perceptual quality.
>
> ### 6. Typos
> We appreciate your attention to detail and will thoroughly proofread the final version to eliminate such issues.

---

> > ### Comment · Reviewer_wa33 · 2025-04-04
> >
> > Thank you for your rebuttal! After reviewing the rebuttal and the thoughts of other authors, I believe that my current rating is fair and I will maintain it (Weak Accept). The paper presents a new coherent framework for adversarial relighting attacks but both myself and the other reviewers maintain that the technical contribution is on the simpler side. The quantitative results are also somewhat mixed. I'm thus still on the fence about the paper but will lean towards acceptance given the clarifications from the rebuttal and the potential for this work to develop further.

---

> > > ### Author Response · Authors · 2025-04-05
> > >
> > > We respectfully appreciate your recommendation and your decision to maintain a positive attitude towards our work.

---

### Official Review · Reviewer_cRHQ · 2025-03-18

**Overall Recommendation:** 4

**Summary:**

This paper presented a relighting-based adversarial attack method against pre-train vision-language models. Given an image, the attack first consult ChatGPT for initial attacking lighting parameters. Based on the parameters, a lighting image is generated using Comfyui-ic-light. Next, IC-Light is used to relight the target image based on the lighting image generated from last step. Finally, the attack image is improved through a two-step optimization approach. Evaluation shows the proposed method can outperform previous non-suspicious adversarial attack methods. The qualitative examples look realistic.

## update after rebuttal

The rebuttal has address all my concerns and added some interesting results.

**Claims And Evidence:**

* Claims: pre-trained VLMs are vulnerable to relighting-based attacks
* Evidence: empirical experimental results on three models, two datasets, and two downstream tasks

**Essential References Not Discussed:**

Did not find.

**Experimental Designs Or Analyses:**

* While it's good to show the impact of two downstream tasks, it's unclear why classification, which is still a popular task for pre-trained VLMs, is not evaluated. Performance degradation is easier to measure for classification.

* The ablation study on ChatGPT shows very small NIQE change in Table 3, yet the qualitative example show in Figure 7 is very different. Why? Is average NIQE not a good metric? Or the example in Figure 7 is an extreme case?

* Are the evaluated models robustness against nature (non-adversarial) relighting? The current results do not rule out the possibility that these models are just bad.

* Is GPT-4 vulnerable to the attack images?

**Methods And Evaluation Criteria:**

Previous studies have shown relighting is a viable approach to generate non-suspicious adversarial attacks. The proposed method incorporate recent advances in relighting image generation (IC-Light) to launch the attack. The effectiveness of most attack steps are evaluated with an ablation study.

Target models, benchmark datasets, and performance metrics are standard. NIQE is used to assess the naturalness of attack images.

My main complaint the lacking baseline performance on clean images.

**Other Comments Or Suggestions:**

None

**Other Strengths And Weaknesses:**

Strength: qualitative examples look very natural yet the task outputs (captioning and VQA) are really bad.

**Questions For Authors:**

1. What's the model's baseline performance on clean images?
2. What's the model's performance on natural but non-adversarial relighting images?
3. Are GPT-4 models vulnerable to the attack images?

**Relation To Broader Scientific Literature:**

Re-confirmed vision models are not robust under different lighting.

**Theoretical Claims:**

N/A

---

> ### Author Rebuttal · Authors · 2025-03-31
>
> ### 1. Baseline performance on clean images
> Thank you for this comment. The baseline performance on clean images for both image captioning and VQA tasks (shown in Table 1 and Table 2 of the original submission) can be found at: https://imgur.com/a/Kk9VYDG. These results confirm that our method induces a significant performance drop across both tasks, highlighting its effectiveness as an adversarial attack.
>
> We will include the clean image results in the final version of the paper, either in the main paper or the appendix, to ensure a clear and direct comparison between clean and adversarial performance.
>
> ### 2. Performance on classification task
> Thank you for your valuable suggestion. While classification is indeed important, we prioritize image captioning and VQA as they pose more complex cross-modal reasoning challenges and provide clearer semantic outputs, making them better suited for evaluating the subtle effects of our non-suspicious adversarial relighting. Additionally, these tasks help avoid label-space ambiguity that can arise in classification evaluations.
>
> We have added a new classification experiment using CLIP on the ImageNet dataset. In this experiment, we compare LightD with three representative non-suspicious color-based attack baselines (ALA, AdvCF, and ColorFool) and report Top-1 and Top-5 classification accuracy as well as NIQE scores for visual quality.
>
> As shown in the following table, LightD causes a significantly larger drop in classification accuracy while achieving better perceptual quality than the baselines. These results demonstrate that LightD remains effective and transferable beyond captioning and VQA and can generalize well to vision-only classification tasks. The full results and analysis will be included in the revised manuscript.
>
> |Model|Top1_acc|Top5_acc|NIQE|
> |-|-|-|-|
> |Clean Image|0.601| 0.842|9.530|
> |ColorFool| 0.183| 0.615|9.884|
> |AdvCF| 0.063|0.404|10.068|
> |ALA| 0.565| 0.838|9.824|
> |**LightD(Ours)**|**0.043**|**0.107**|**8.429**|
>
> ### 3. NIQE result analysis
> Thank you for your insightful question. NIQE is a widely used no-reference metric, but it is known to be neither perceptually linear nor highly sensitive to subtle semantic variations. As such, small numerical differences in NIQE may correspond to perceptually noticeable differences, particularly in edge cases. Therefore, we also provide qualitative examples (e.g., Figure 7) to complement the quantitative evaluation.
>
> In Table 3, the average NIQE difference between randomly selected lighting parameters and ChatGPT-recommended ones is relatively small, indicating that both strategies generally yield visually acceptable adversarial relighted images. However, the example in Figure 7 was selected to highlight a representative case where ChatGPT’s lighting suggestions lead to better semantic consistency and perceptual quality, especially in preserving scene structure and object boundaries. These refinements are often more apparent to human observers than reflected by NIQE scores.
>
> ### 4. Performance on natural (non-adversarial) relighting
> Thank you for this comment. We conduct comparable experiments to test the robustness against natural relighting. Specifically, we use IC-Light to generate the natural lighting effects simulating day and night environments without any adversarial optimization. As shown in the following table, these natural relightings (Natural-Day and Natural-Night) induce only a minor degradation in performance compared to clean images. In contrast, LightD leads to a much more significant drop across all captioning metrics, demonstrating that the observed attack effectiveness is not merely due to lighting variation but rather to carefully optimized adversarial perturbations.
>
> | **Attack** | **BLEU↓** | **METEOR↓** | **ROUGE↓** | **CIDEr↓** | **SPICE↓** | **NIQE↓** |
> |---|---|----|--|--|--|--|
> | Clean Image| 0.738 | 0.267 | 0.539  | 1.094  | 0.198      | 9.530     |
> | Natural-Day| 0.728  | 0.257   | 0.540      | 1.082  | 0.190      | 11.453    |
> | Natural-night | 0.704     | 0.243       | 0.519      | 0.964      | 0.173      | 10.879    |
> | **LightD (Ours)**| **0.460** | **0.119**   | **0.340**  | **0.211**  | **0.055**  | **8.650** |
>
> ### 5. Is GPT-4 vulnerable to the attack images?
> Thank you for the question. Although our method is designed as a white-box attack, we further evaluate its transferability to GPT-4. Specifically, we use BLIP-2 as the surrogate model on the MSCOCO dataset for the VQA task and test the performance of the generated adversarial images on GPT-4.
>
> As shown in the following table, the adversarial examples cause a drop in both APA and WUPS0.9 scores on GPT-4, demonstrating that our method retains a certain degree of transferability and attack effectiveness even against large, black-box models like GPT-4.
>
> | Model  | **APA↓** | **WUPS0.9↓**|
> |--------|----------|-------------|
> | Clean Image|32.48 |34.07  |
> | Adv Image  |28.02 |27.11   |

---

### Decision · Program_Chairs · 2025-05-01

**Decision:**

Reject

**Comment:**

The paper introduces LightD, a method for generating natural adversarial examples against vision‐and‐language pretraining (VLP) models by leveraging semantically guided relighting. The approach first uses ChatGPT to propose context‐aware lighting parameters, then applies a pretrained relighting model (IC-Light) to produce a reference lighting image. This initial output is further refined using gradient-based optimization to enhance the adversarial effectiveness. Extensive experiments across multiple VLP models, datasets, and tasks are provided to demonstrate the method’s efficacy.

Overall, the reviewers find the problem setting here is new, the quantitative evaluations are strong, and the qualitative visualization looks natural. But meanwhile, some major concerns are raised: 1) while the overall application is new, much of the method is a straightforward integration of existing tools; 2) the baselines used for comparison are relatively dated; 3) certain results needs further analysis (e.g., NIQE score); 4) the computational cost introduced by the two-step optimization requires more analysis; 5) additional evaluations are needed (e.g., transferability, robustness against defense).

The rebuttal is considered, but fail to fully address all these concerns, especially regarding its limited technical contribution (which remains a significant concern raised by Reviewer 3ss3 and echoed in comments from Reviewers j7H2 and wa33). While the problem setting is intriguing, the AC concurs that the paper lacks substantial technical innovation. Consequently, the final decision is rejection.